

# Coastal Sea Level rise at Senetosa (Corsica) during the Jason altimetry missions

Yvan Gouzenes[1], Fabien Léger[1], Anny Cazenave[1,2], Florence Birol[1], Pascal Bonnefond[3], Marcello Passaro[4], Fernando Nino[1], Rafael Almar[1], Olivier Laurain[5], Christian Schwatke[4], Jean-François Legeais[6] and Jérôme Benveniste[7]

1.LEGOS, Toulouse; 2. ISSI, Bern; 3. Observatoire de Paris-SYRTE, Paris ; 4. TUM, Munich; 5. Observatoire de la Côte d'Azur-Géoazur, Sophia-Antipolis; 6. CLS, Ramonville St Agne; 7. ESA-ESRIN, Frascati.

**3rd Revision**

**9 August 2020**

Corresponding author: Anny Cazenave (anny.cazenave@legos.obs-mip.fr; anny.cazenave@gmail.com)

## **Abstract**

In the context of the ESA Climate Change Initiative project, we are engaged in a regional reprocessing of high-resolution (20 Hz) altimetry data of the classical missions in a number of world's coastal zones. It is done using the ALES (Adaptive Leading Edge Subwaveform) retracker combined with the X-TRACK system dedicated to improve geophysical corrections at the coast. Using the Jason-1&2 satellite data, high-resolution, along-track sea level time series have been generated and coastal sea level trends have been computed over a 14-year time span (from July 2002 to June 2016). In this paper, we focus on a particular coastal site where the Jason track crosses land, Senetosa, located south of Corsica in the Mediterranean Sea, for two reasons: (1) the rate of sea level rise estimated in this project increases significantly in the last 4-5 km to the coast, compared to what is observed further offshore, and (2) Senetosa is the calibration site for the Topex/Poseidon and Jason altimetry missions, equipped for that purpose with in situ instrumentation, in particular tide gauges and GNSS antennas. A careful examination of all the potential errors that could explain the increased rate of sea level rise close to the coast (e.g., spurious trends in the geophysical corrections, imperfect intermission bias estimate, decrease of valid data close to the coast and errors in waveform retracking) has been carried out, but none of these effects appear able to explain the trend increase. We further explored the possibility it results from real physical processes. Change in wave conditions was investigated but wave set up was excluded as a potential contributor because of too small magnitude and too localized in the immediate vicinity of the shoreline. Preliminary model-based investigation about the contribution of coastal currents indicates that it could be a plausible explanation of the observed change in sea level trend close to the coast.


## 1. Introduction

Since the early 1990s, satellite altimetry provides invaluable observations of the global mean
sea level and its regional variability. In the recent years, this data set has generated an
abundant literature on the processes causing sea level change at global and regional scales, as
well as on closure of the sea level budget (e.g., Church et al., 2013, Stammer et al., 2013,
Dieng et al., 2017, Nerem et al., 2018, WCRP, 2018, SROCC, 2019). In addition to the global
mean rise and superimposed regional trends, changes in small scale processes such as local
atmospheric effects, baroclinic instabilities, coastal trapped waves, shelf currents, waves,
fresh water input from rivers in estuaries, can substantially modify the rate of sea level change
at the coast compared to open sea regions (Woodworth et al., 2019, Melet et al., 2018,
Piecuch et al., 2018, Dodet et al., 2019, Durand et al., 2019). In addition, ground subsidence
may amplify the rate of sea level change at the coast (Woppelmann and Marcos, 2016). In
terms of societal impacts, what really matters in the coastal zone is indeed the sum of the
global mean sea level rise plus the regional trends and the local processes.
Up to recently, due to land contamination of radar echoes and less precise geophysical
corrections, classical altimetry did not provide reliable sea level data in a band of 10-15 km
along coastlines. However different studies have shown that using adapted reprocessing of
altimetry measurements and improving geophysical corrections allows retrieving a large
amount of valid sea level close to the coast (e.g., Cipollini et al., 2018, Passaro et al., 2015,
Marti et al., 2019). In addition, despite having a much higher noise level than the classical 1
Hz altimetry data, high-resolution 20 Hz measurements allow to recover more information on
coastal sea level variations (Birol and Delebecque, 2014, Leger et al., 2019).
In the context of the Climate Change Initiative (CCI) project of the European Space
Agency (ESA), we have initiated a reprocessing of high-resolution (20 Hz) altimetry data of
the Jason-1 and Jason-2 missions along coastal zones of Western Africa, Northern Europe and
Mediterranean Sea. The ALES (Adaptive Leading Edge Subwaveform) retracker (Passaro et
al., 2014) was applied to estimate the satellite-sea surface distance (called range) which was
further combined with the X-TRACK processing chain dedicated to improve geophysical
corrections at the coast (Birol et al., 2017). This allowed us to derive along-track sea level
anomaly (SLA) time series (Leger et al., 2019) from which coastal sea level trends were
estimated. Results show that in a number of sites, coastal sea level rates computed over a 14-
year time span (2002-2016) significantly deviate from the open ocean rate within 5 km to the
coast (Marti et al., 2019).

In the present study, we focus on a particular site, Senetosa, located southern Corsica in the Mediterranean Sea (41° 33'N, 8°48'E), for two reasons: (1) in this region, the computed rate of sea level rise increases significantly in the last 3-5 km to the coast, and (2) there is a Jason satellite track that crosses land at Senetosa, a calibration site for altimetry missions chosen since the launch of the Topex/Poseidon mission in 1992 and equipped for that purpose with in situ instrumentation, in particular tide gauges and GNSS antennas (Bonnefond et al., 2019). This calibration site provides an independent reference to explore the near-shelf signal observed in altimetry data.

## 2. Data and method

As presented in detail in Marti et al. (2019) and Léger et al. (2019), here we use the regional X-TRACK/ALES along-track 20 Hz SLA data derived from Jason-1 and Jason-2 missions (DOI: 10.5270/esa-sl_cci-xtrack_ales_sla-200201_201610-v1.0-201910). This product is based on new ranges and new sea state bias corrections estimated using the ALES retracker (see details on the retracking methodology in Passaro et al., 2014), and further combined with the X-TRACK software developed at CTOH (Center of Topography of the Ocean and the Hydrosphere) at LEGOS (Laboratoire d'Études en Géophysique et Océanographie Spatiales). The new X-TRACK/ALES processing system first downloads from the altimetry database hosted by the French National Observations Service for altimetry called CTOH (http://ctoh.legos.obs-mip.fr/), all parameters needed to compute the sea level anomaly (orbit solution, altimeter ranges, instrumental, environmental and geophysical corrections). These parameters come from the Geophysical Data Records (GDRs) data sets distributed by the space agencies for the different altimetry missions. ALES range and SSB products come from TUM. Additional geophysical corrections are provided by the RADS altimeter database (http://rads.tudelft.nl/rads/rads.shtml) and the University of Porto (for the GPD+ wet tropospheric correction, Fernandes et al., 2015). Concerning the geophysical corrections, we used the standards defined in the ESA CCI sea level project (http://www.esa-sealevel-cci.org/). These are summarized in Table 1.

| Parameter | Source | Jason-1 / Jason-2 |
|-----------|--------|-------------------|
| Altitude | GDR | Altitude of satellite |
| Range | ALES/TUM | 20 Hz Ku band ALES corrected altimeter range (Passaro et al. 2014) |

| Sigma0 | ALES/TUM | 20 Hz Ku band ALES altimeter sigma0 (Passaro et al. 2014) |
|---|---|---|
| Ionosphere | GDR | From dual-frequency altimeter range measurement |
| Dry troposphere | GDR | From ECMWF model |
| Wet troposphere | University of Porto | GPD+ correction (Fernandes et al. 2015) |
| Sea-state bias | ALES/TUM | Sea-state biais correction in Ku band, ALES retracking (Passaro et al. 2018) |
| Solid tides | RADS | From tide potential model (Cartwright and Taylor 1971, Cartwright and Eden 1973) |
| Pole tides | GDR | From Wahr 1985 |
| Loading effect | RADS | From FES 2014 (Carrere et al. 2012) |
| Atmospheric correction | RADS | From MOG2D-G (Carrere and Lyard 2003) + inverse barometer |
| Ocean tide | RADS | From FES 2014 (Carrere et al. 2012) |


*Table 1: List of altimetry parameters and geophysical corrections used in the computation of the*
*coastal sea level products.*

A dedicated editing strategy was further applied to eliminate noisy data. For each orbit cycle, the
temporal behavior of each geophysical correction was analyzed along the satellite track. Abrupt
changes were considered as spurious and removed (Birol el al., 2017). This strategy has proved to
be very efficient in recovering a significant amount of valid altimeter measurements that were
otherwise flagged in the standard GDR products (Jebri et al., 2016). In a second step, all corrections
were recomputed at the 20-Hz high-rate using only the valid data, through interpolation/extrapolation
method. The sea level data of each cycle were further projected onto fixed points along a nominal
ground track and converted into SLAs by subtracting a reference mean sea surface. At this stage of
the processing, a regional dataset of SLA time series with a spatio-temporal resolution of 10 days
and 20Hz (~0.3 km) was produced for each Jason mission. To obtain a single multi-mission product,
an inter-mission bias was  estimated and removed. This was done at regional level by computing the
mean sea level differences between the two missions over their overlapping period (calibration
phase). The resulting SLAs were further averaged on a monthly basis at every 20 Hz point and an
additional editing was performed to remove outliers (details in Marti et al., 2019).
In this study we focus on the section of Jason track 85 located off the southwestern coast of
Corsica island (Western Mediterranean Sea) (see Fig. 1).

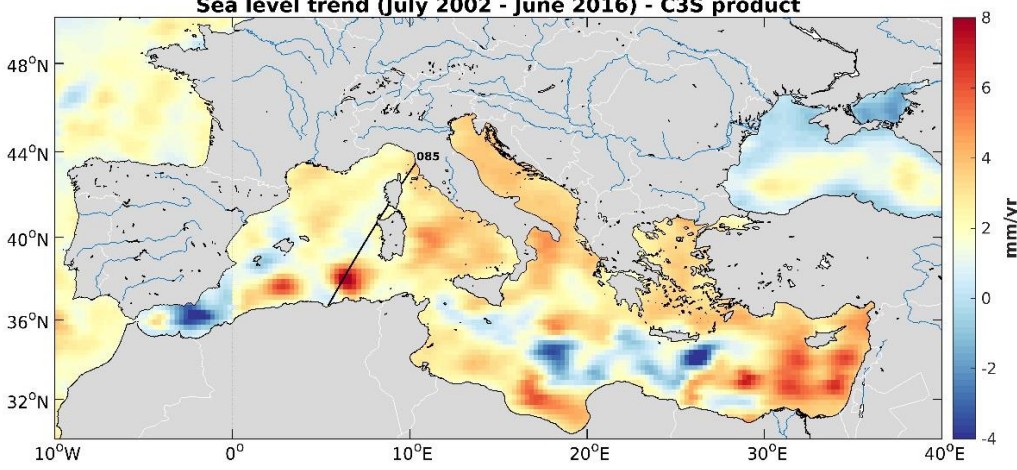



*Fig. 1: Location of the Jason track 85 crossing Corsica at the Senetosa site (black straight line).*
*The background maps shows sea level trends over 2002-2016, based on gridded altimetry data*
*from the Copernicus Climate Change Service (C3S)*

## 3. The Senetosa calibration site

Since 1998, a calibration site of the Topex/Poseidon and Jason missions has
operated near the Senetosa lighthouse with support from CNES (Centre National
d'Études Spatiales, France), NASA (National Aeronautics and Space Administration, USA)
and the Observatoire de la Côte d'Azur (France). It is equipped with different in situ
instrumentation, including weather stations, several tide gauges and GNSS antenna. Since
1998, this calibration site has been widely used to validate the altimetry-based sea surface
height data (Bonnefond et al., 2003a,b, 2010, 2011). Fig.2 is a Google Earth image of the
coast, showing the geographical configuration of the Senetosa calibration site, with the
location of the tide gauges, the GNSS antenna and the Jason track. Three tide gauges were
operating during our study period (M3, M4 and M5). M4 and M5, a few tens of cm apart,
are located on the western part of the coastline sheltered from northwestward wind
forcing. M3 at 1.7 km eastward of M4/M5 is more exposed to open sea conditions from the
west.
Vertical land motion time series are available from the GNSS reference receiver located close
to the lighthouse (G0 reference marker in Fig.2). The tide gauges have been regularly leveled
relatively to the G0 reference marker with no relative motion detected so far at the millimeter
level over 10 years. Trends in sea level and vertical land motions derived from these
instruments at Senetosa are discussed in section 5.

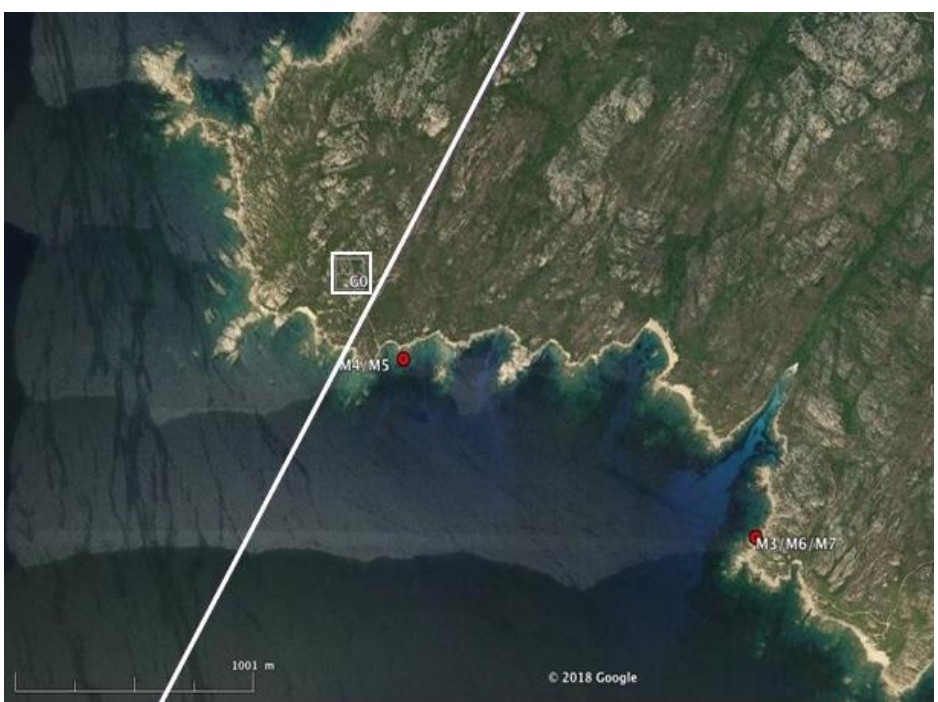


*Fig. 2: Google Earth image of the Senetosa calibration site. The two tide gauge sites (referred*
*as M4/M5 and M3) are shown by the red dots. The G0 reference marker (G0) is indicated by a*
*white square and the Jason ground track by the white straight line.*

## 4. Analysis of the coastal sea level trends off Senetosa

### 4.1 Coastal sea level trends derived from altimetry data

Following the data processing described above, we focus on monthly SLA time series sampled

at 20 Hz (~350 m in the along-track direction), from 15 km offshore to the coastline. Examples

of along-track SLA time series at coastal points, located at 1 km, 1.6 km, 2.2 km, 5 km and

15 km from the coast respectively, are shown in Fig.3.

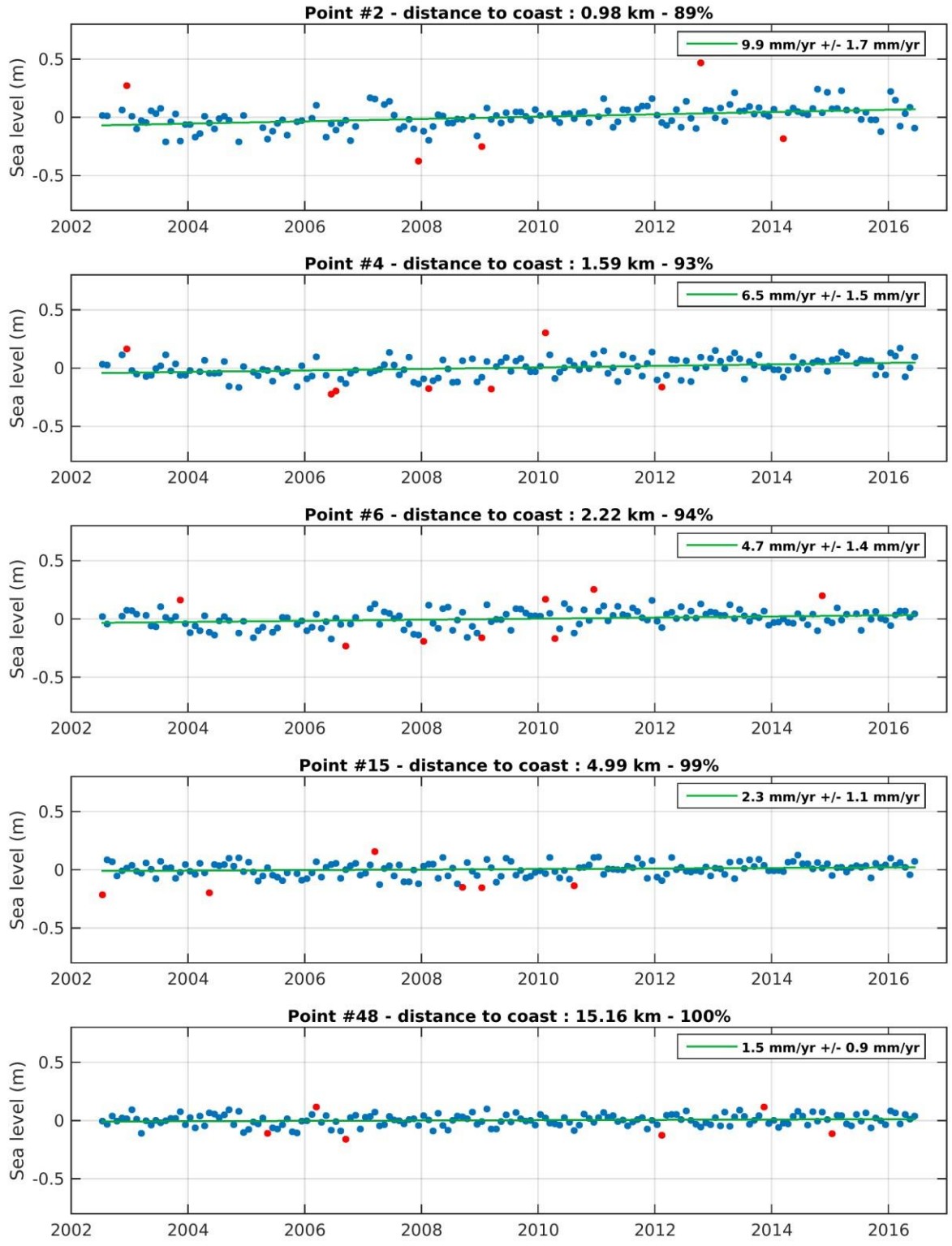

*Fig. 3: Examples of sea level anomalies time series for 20-Hz points located at different distances from the coast. The distance to coast, percentage of valid data and sea level trends are indicated on each plot. The green curve is the regression line adjusted to the data. The red points on the time series correspond to outliers detected using a simple 2-sigma filter (sigma corresponding to the SLA standard deviation). These are not considered to compute the regression line.*


For each 20 Hz point, we have then computed the regression line of the resulting SLA time
series and the associated standard deviation (1-sigma) based on the least squares fit, to estimate
sea level trends over the  study time span. Fig.4 shows the corresponding along track sea level
trends as a function of  distance to the coast (from 15 km offshore).

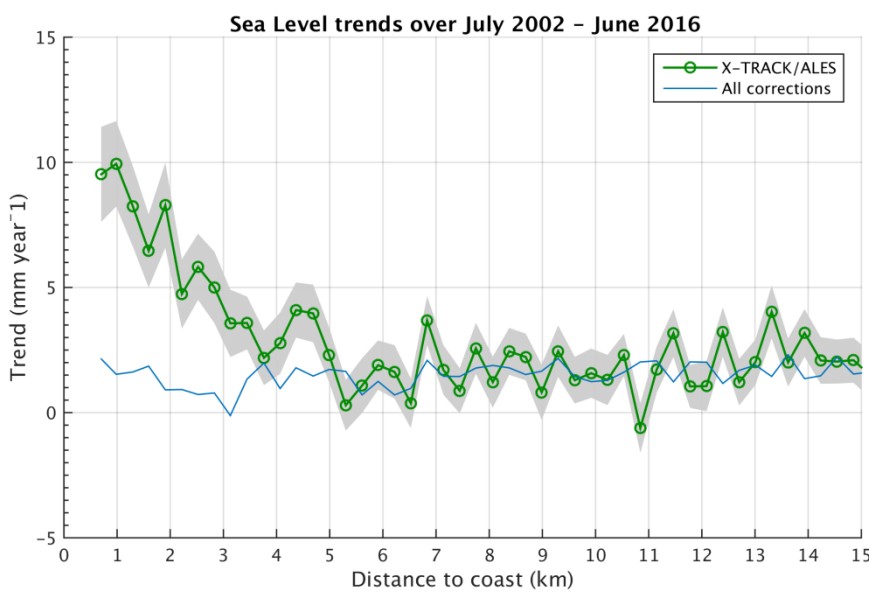


*Fig. 4: Altimetry-based sea level trends over July 2002-June 2016 around Senetosa as a*
*function of distance to the coast .  Shaded area corresponds to trend uncertainty range. The*
*light blue curve is the sum of trends  in individual corrections.*

As shown in Fig.4,  beyond ~ 5 km from the coast towards the open sea, the trend over 2002-
2016 is relatively stable and on average on the order of 2-3 mm/yr. High frequency
oscillations around this value are observed between adjacent points but these are likely due to
noise and we note they are of the same order of magnitude or only slightly larger than the
standard deviation of trend estimates at each point (of ~1.5 mm/yr).
As also shown in Fig.4, we note an almost continuous increase in the trend in the last ~4-5 km
to coast. The  corresponding trend uncertainties (standard deviation)  are  not  significantly
larger than  offshore (<2 mm/yr).

## 4.2 Robustness of the computed coastal trends


In coastal areas, precision of sea surface height from altimetry is limited by inaccuracies in


some of the applied geophysical corrections (including sea-state bias, wet tropospheric


correction, dynamical atmospheric correction and ocean tides) and from the distorted shape of


the radar waveforms as the satellite approaches land (Vignudelli et al., 2011 and Cipollini et al.,


2018).


The corresponding altimetry measurements are often discarded by the processing chains or


flagged in the data sets as potentially erroneous, leading to low confidence sea level


trend estimates near the coastline. These estimates can also be impacted by the lower


percentage of valid data in the coastal zone, as well as by the uncertainty in the bias estimate


between the two successive missions Jason-1 and Jason-2. In order to check whether the sea


level trend increase close to the coast reported in section 4.1 is associated to one of these


factors, each of them is independently examined.



### 4.2.1 Coastal errors in the geophysical corrections


We first computed and plotted the geophysical correction trends against distance to the coast


for the sea-state bias (ssb), wet atmospheric correction, atmospheric loading (called DAC-


dynamic atmospheric correction-) and ocean and loading tide correction (Fig.5).



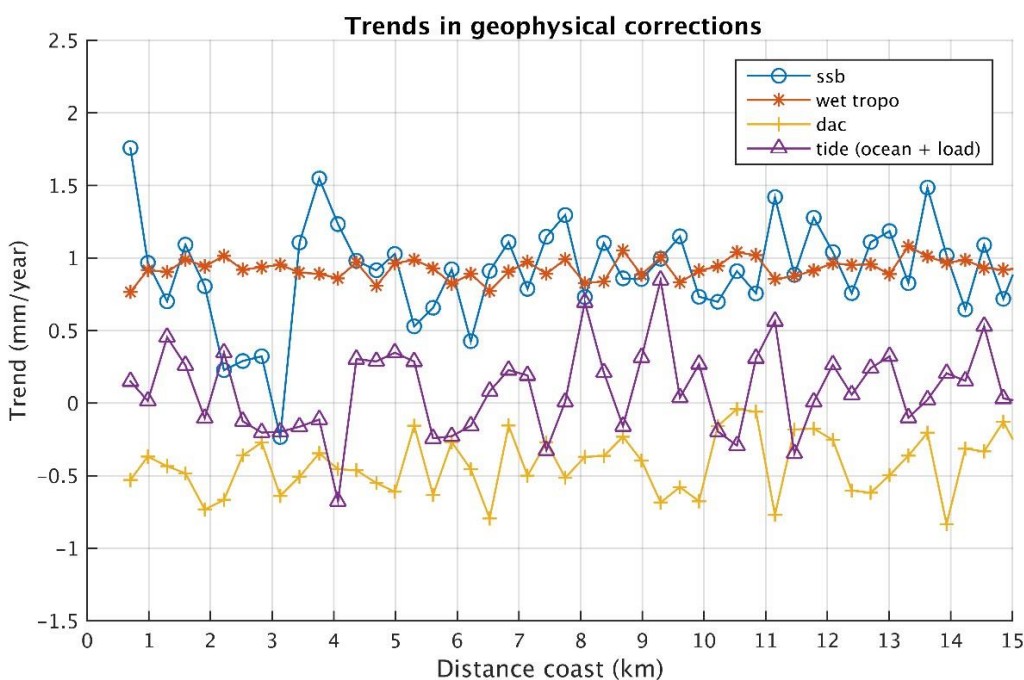



*Fig. 5: Trends in the geophysical corrections (sea state bias/ssb, wet tropospheric correction,*
*dynamic atmospheric correction/dac, ocean tide plus ocean loading tide) as a function of*
*distance to coast. Note that the vertical scale is different from Fig.4.*

Trends in the geophysical corrections are rather small and their amplitude in the range +/- 1
mm/yr, except for the ssb that shows a larger trend within 4 km to coast, but always less than
2 mm/yr. It is worth mentioning that the ssb is a function of significant wave height (SWH)
and backscatter coefficient (both related to wind speed). In the ALES retracking the ssb is
recomputed for each 20-Hz point. So a trend in ssb may be due to either a different behavior
of the SWH and wind speed at the coast, or to changes in backscatter properties.
The sum of these geophysical correction trends is plotted in Fig.4 (blue line). Even if the
geophysical corrections, and especially the ssb, are more uncertain close to the coast, Fig. 4
suggests that the continuous increase in the sea level trends observed in the last ~4 km to the
coast may not be due to trends in the geophysical corrections. It remains that the empirical
formulation used for the ssb correction may not be valid close to the coast where waves could
have a different behavior compared to the open sea. This will be discussed in section 6.1.

*4.2.2 Coastal changes in the percentage of valid data*
We next examined the possible impact on the trend estimation of the decrease in valid data in
the last 3-4 km to coast. The original percentage of valid data at each 20-Hz point decreases
with distance to the coast, as shown in Fig.6. We resampled the along-track sea-level records
keeping only the 80% of data common to all along track positions at a given time.


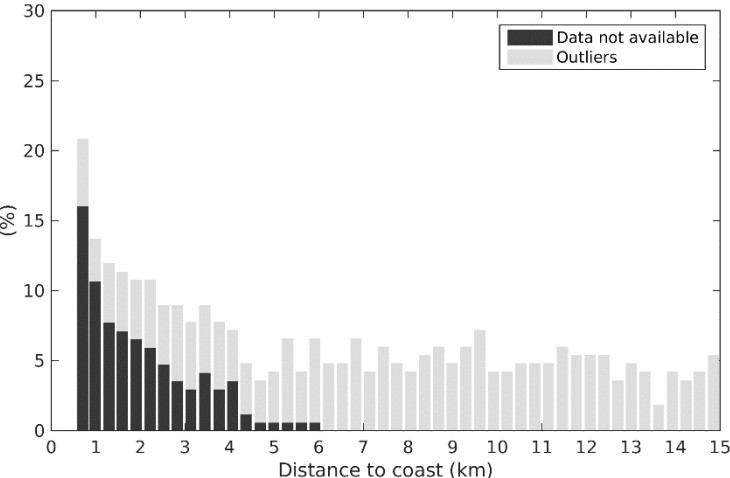

*Fig. 6: Percentage of missing points for the original data set.*

The along-track sea level trends were recomputed with the new sampling (80% of the original
data kept) (Fig.7). For comparison, in Fig.7 we superimpose the trends computed with the
original sampling. Trends compare well in both cases. Even if the trend values are
slightly lower in the band 0-5 km, keeping only 80% of the valid data does not change
significantly the coastal trend behavior. We conclude that the lower amount of valid near-
shore altimetry data does not explain the trend increase observed as the distance to the coast
decreases.

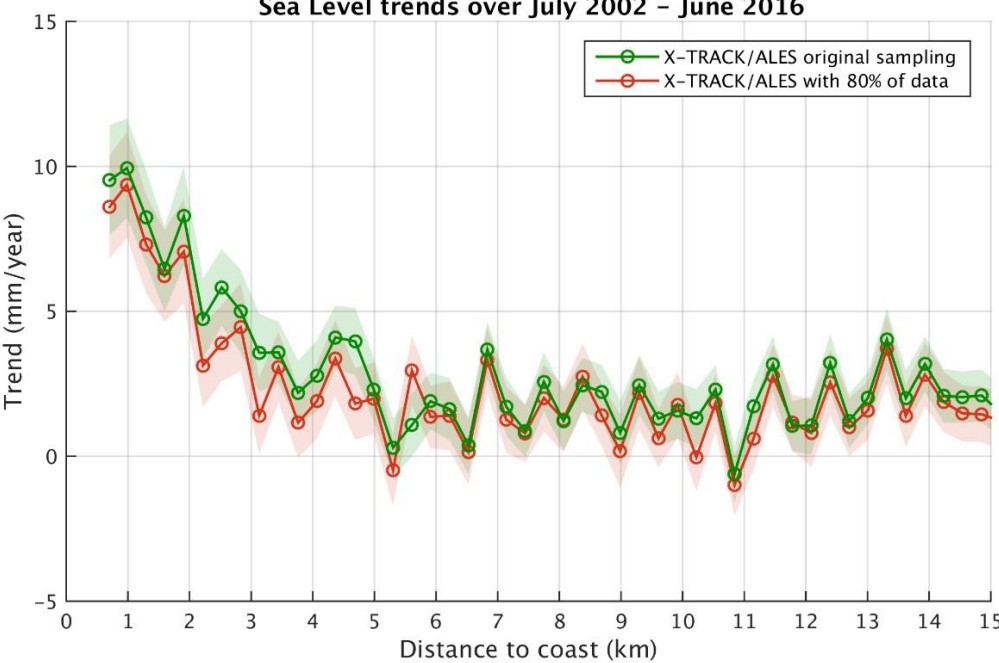



*Fig. 7: Sea level trends as a function of distance to the coast with the original data set (green*
*curve) and new sampling (80% of original data kept; red curve).*

*4.2.3 Effect of intermission bias estimation*
As discussed in detail in Marti et al. (2019), in the X-TRACK/ALES sea level product, the
bias applied to combine the Jason-1 and Jason-2 data in a single sea level time series was
estimated at a regional scale. In the case of our study region, it was estimated over the whole
Mediterranean Sea. In order to investigate a possible impact of this approach on the sea level
trend estimates, we tested other bias calculation methods. We first recomputed the
intermission bias along the Jason track 85 (using only measurements of this particular track).
In another test, the bias was computed from data included in a 1x1 degree box around the
Senetosa site. The sea level trends derived from the corresponding Jason-1 and Jason-2 time
series are shown in Fig. 8a for these two cases, superimposed to the regional bias case shown
in section 4.1. Here again, we can see that there is almost no difference between the results of
the three approaches, indicating that inadequate intermission bias estimate does not explain
the coastal trend increase. To complete these tests, we also recomputed SLA trends as a
function of distance to coast using as reference a local geoid computed for altimetry mission
calibration purposes (P. Bonnefond, personal communication). Fig.8b shows the geoid profile
together with the along-track mean sea surface computed with the altimetry data, as a function
of latitude. Both references compare well Thus, as expected, exactly the same trend increase
behavior as a function of distance to coast is observed when the reference geoid is used
(figure not shown as it is similar to Fig.4). We conclude that the reference has no impact on
the computed trends.

*(a)*

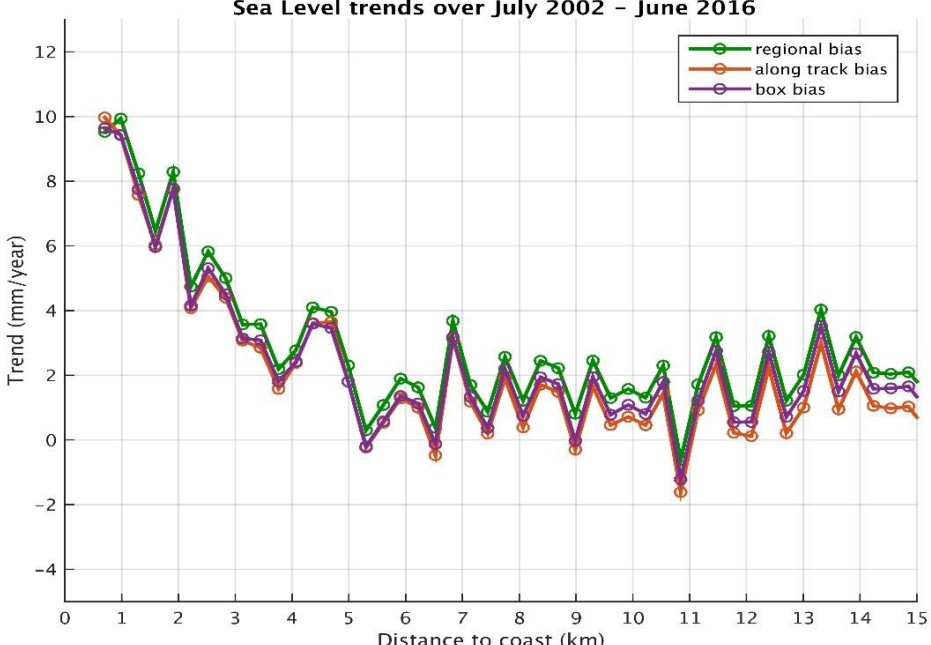


*(b)*

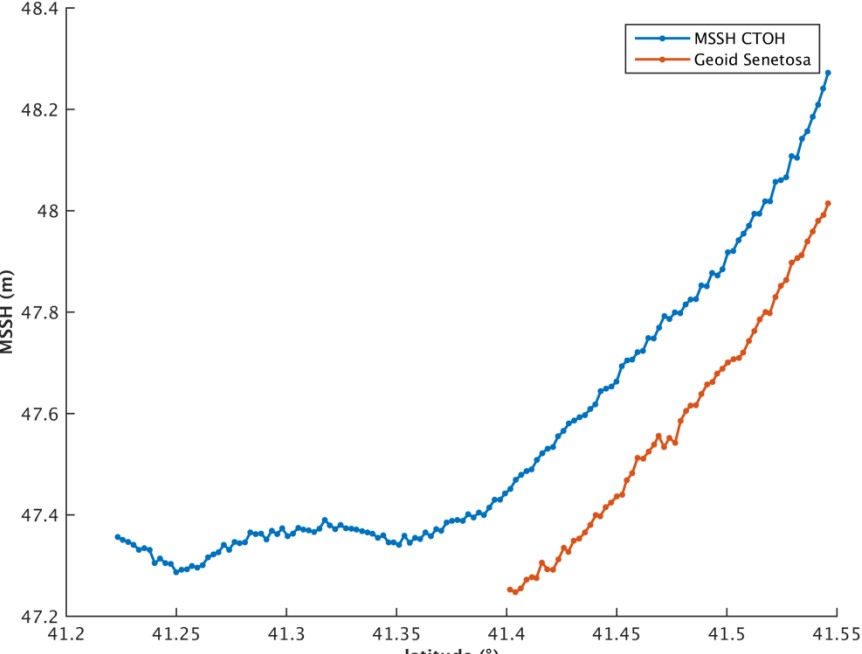


*Fig.8: (a) Sea level trends as a function of distance to the coast for three different intermission*
*bias estimates. (b) Geoid and altimetry-based along-track mean sea surface profiles as a*
*function of latitude.*

*4.2.4 Coastal altimetry waveforms and range values near Senetosa*
In another series of tests, we examined the shape of the radar waveforms at 20 Hz points as a
function of distance to coast, considering a few Jason cycles taken at random. An example is
shown in Fig. 9 for a point located between the coast and 2 km offshore. Fig.9 shows that at
the Senetosa site, the leading edge of the coastal radar echo is generally well defined,
suggesting that a robust determination of the range is possible very close to the coast.

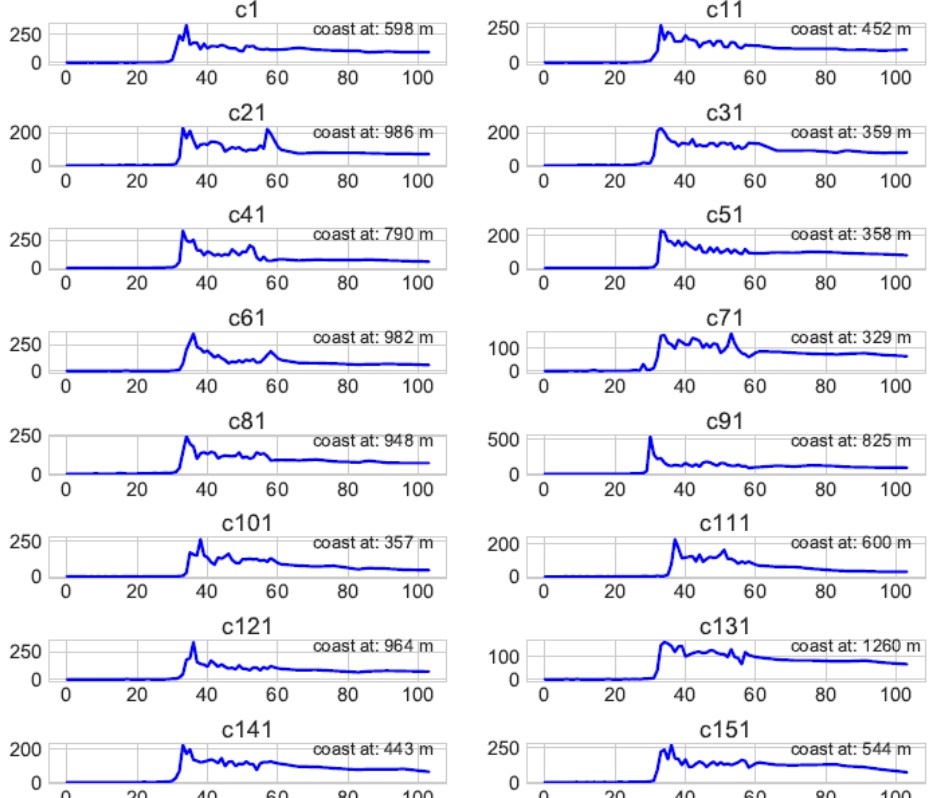



*Fig. 9: Observed radar waveforms at points close to the coast for a series of Jason cycles*
*(numbers on each plot refer to cycle number).*

To investigate this further, we tried to assess the reliability of successive 20-Hz ALES-based
range data very close to the coast. The waveform amplitude represents the radar power as a

function of time. For Jason-2, time is discretized into 104 successive 'gates'. Knowledge of the orbit and radar footprint allows by simple geometric analysis to associate a point on ground (pixel) to a given gate. A numerical simulation has been performed for that purpose (assuming flat land) in order to produce range maps for the Jason track 85, with the goal of precisely locating the point on ground corresponding to the measured waveform. This is illustrated on Fig. 10a and Fig. 10b, showing the geographical configuration and associated radar waveforms for two range measurements located at 0.53 km and 1.4 km distance from coast. The range measurement deduced from the waveform corresponds to the center of the circle representing the radar footprint on the range map.

(a)

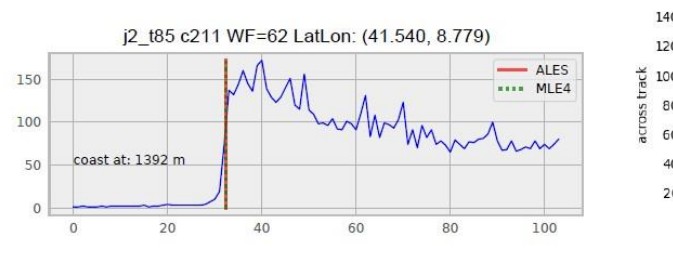 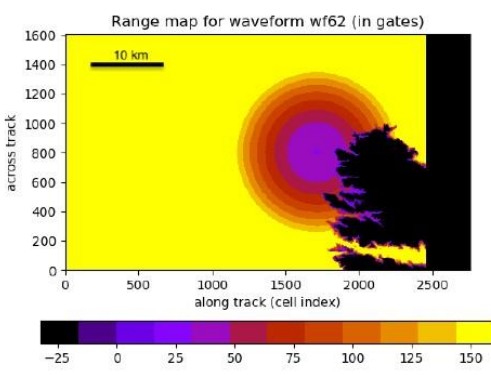

(b)

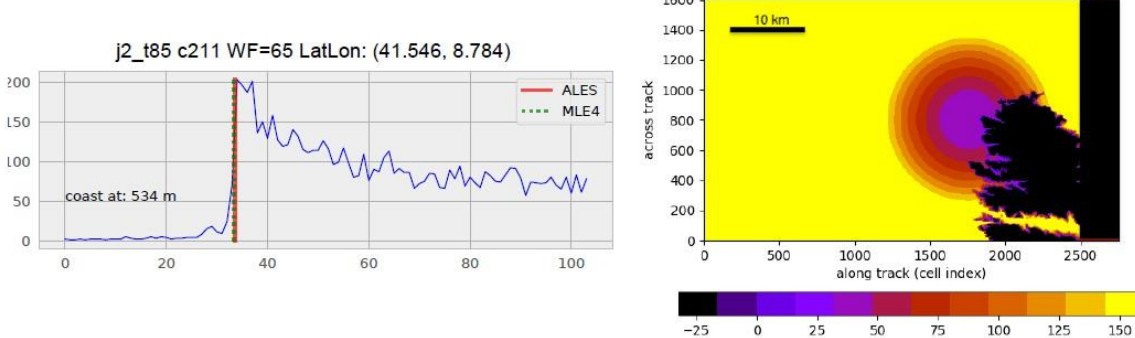

*Fig. 10: (a) Radar waveform as a function of gate number (left) and configuration of the radar footprint on ground (right) at 1.4 km from coast. (b) Same as (a) at 0.5 km from coast.*

Although these simulations represent an ideal case of smooth sea state and flat land, Fig. 10a,b shows that even at the closest point to coast (0.5 km), the leading edge of the return


waveform still corresponds to a reflection of the radar signal on water. This suggests that it is
theoretically possible to retrieve valid sea level information up to 0.5 km to the coast. One
may argue that because the land at Senetosa has some elevation, the real radar echo is partly
contaminated by land reflection at distances larger than the theoretical footprint, even if there
is no wave. However, considering that the real waveform has a leading edge, and t h a t
the  retracker is able to follow it, we conclude that the trends reported on successive 20-Hz
points  are not spurious. Besides, if the retracker was corrupted by inhomogeneous backscatter
properties within the satellite footprint, these should be random (e.g., Passaro et al. 2014).
Finally, 20-Hz waveforms being independent samples, if the retracker is wrong and produces
spurious trends, the latter also would be random. Thus, we should not see a continuous trend
increase over several consecutive points.

*4.2.6 Comparison between ALES  and MLE4 retrackers*
Finally, we performed the same analysis (computation of sea level trends as a function of
distance to the coast) using SLA data computed with the classical MLE4 retracker (used for
the          standard          Geophysical          Data          Records          production,
https://www.aviso.altimetry.fr/fileadmin/documents/data/tools/hdbk_tp_gdrm.pdf).          MLE4-
based trends over the 14-year time span are shown in Fig. 11, on which are superimposed the
ALES-based trends, for comparison. We note that MLE4 gives noisier results than ALES,
especially at distances less than ~5 km to the coast, but the increase in trends in the last ~4-5
km to the coast is still well visible. This clearly means that the trend increase is not an artifact
due to the use of the ALES retracker.

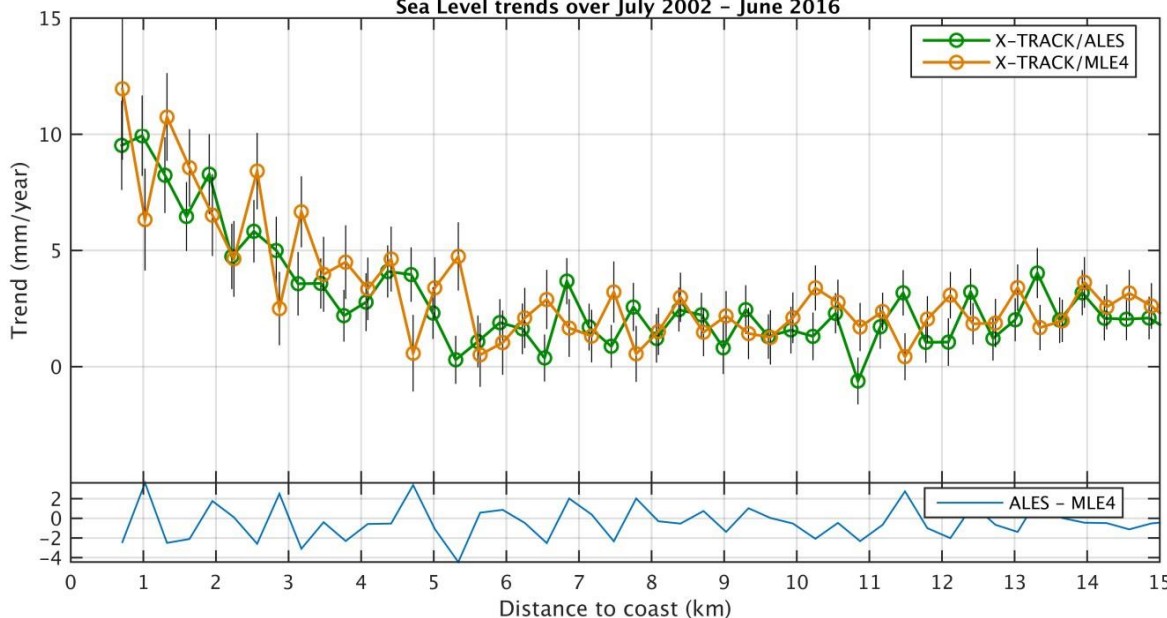


*Fig. 11: Sea level trends as a function of distance to the coast for MLE4 (orange dots) and*
*ALES (green dots)-based SLA data. Vertical bars correspond to trend errors (1-sigma). The*
*light blue curve at the bottom of the panel represents the difference between ALES-based and*
*MLE4-based trends.*

To summarize, from all the tests presented above, we can conclude that the increase in
altimetry sea level trend observed in the last 4-5 km to the coast is not correlated with errors
in the geophysical corrections, is not explained by the loss of valid data, nor the presence of
spurious waveforms or by the intermission bias. Furthermore, the calculated trends are robust
to change in retracker, since instead of using ALES, we also used the standard high-frequency
MLE4 retracker. The corresponding time series still show the same trend behavior (although
with noisier results).

**5. Comparison with the sea level trend derived from tide gauges records**
It is very classical to validate altimetry-based sea level data by comparing with tide gauge
records. The availability of tide gauge records at the Senetosa site is a good opportunity to do
so. Tide gauge data have been provided by the Observatoire de la Côte d'Azur (Géoazur
laboratory) and downloaded from www.aviso.altimetry/fr/en/data/calval/in-situ/absolute-
calibration/download-tide-gauge-data.html. The high-frequency tidal signal and the
atmospheric forcing effect have been removed (using the same DAC correction as for the

altimetry data). The time series have been further smoothed on a monthly basis. The
corresponding tide gauge time series over 2002-2016, for the M3, M4 and M5 tide gauges, are
shown in Fig. 12a and 12b, with and without the seasonal cycles.

(a)

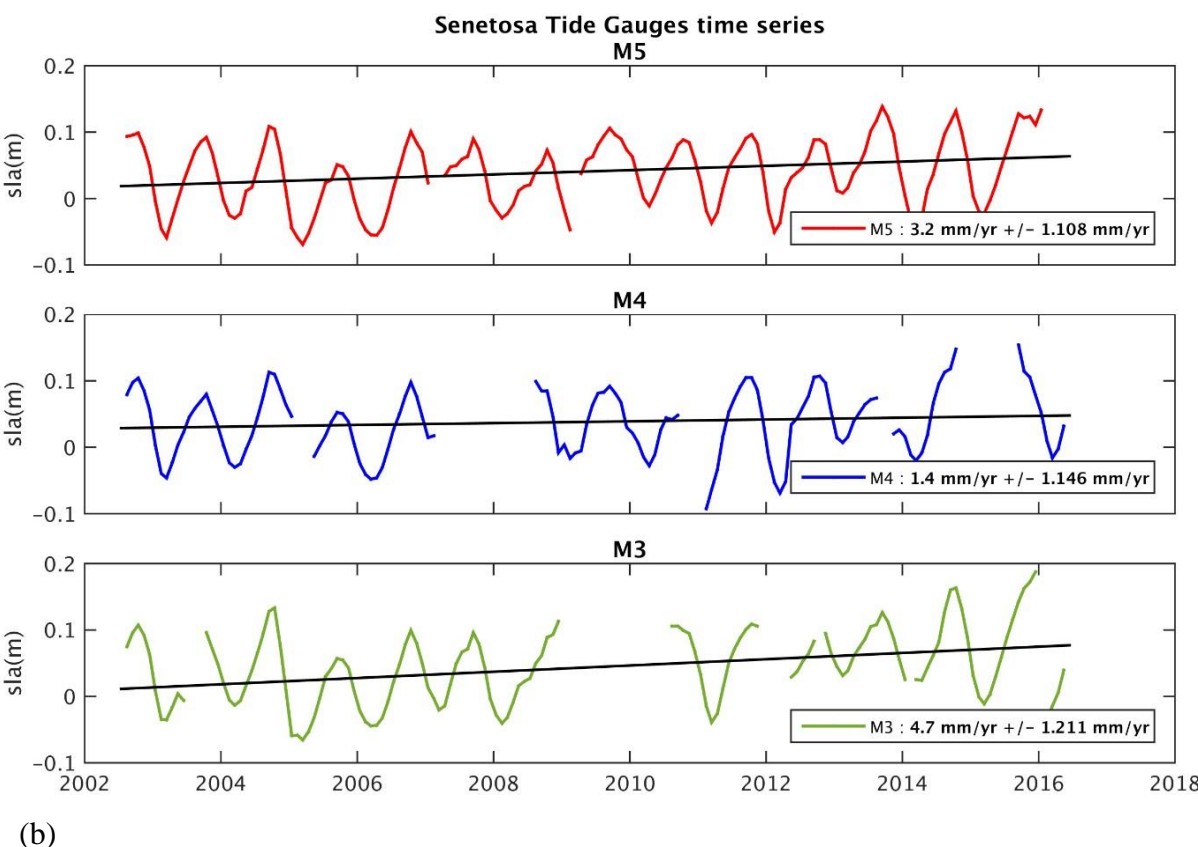


(b)

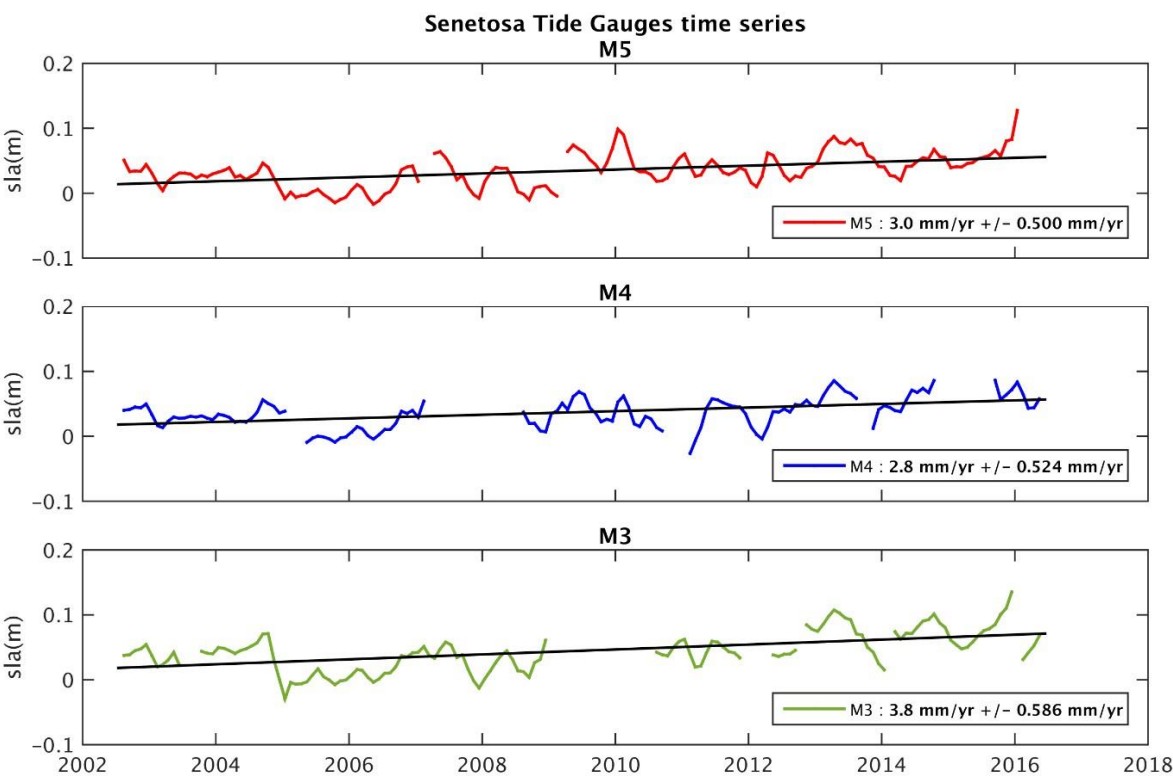


*Fig. 12: Sea level time series based on in situ tide gauges measurements at the M3, M4 and M5 sites over 2002-2016. (a) With the seasonal cycle. (b) Without the seasonal cycle.*

From these time series, we computed linear trends over the same period as for the altimetry data. These are gathered in Table 1 for the two cases (with and without the seasonal cycle). In Bonnefond et al. (2019), it was shown that when making differences between tide gauges sea level measurements, there is no systematic trend between the tide gauge time series since 2001 (below 0.1 mm/yr), well within the trend uncertainties. The GNSS-based vertical land motion (VLM) at Senetosa (estimated in Bonnefond et al., 2019) is also shown. VLM is small at Senetosa, less than 0.3 mm/yr.

| Tide Gauge | Tide gauge trend (mm/yr) (with seasonal cycles) | Tide gauge trend (mm/yr) without seasonal cycles | GNSS VLM (2003-present) (mm/yr) |
|---|---|---|---|
| M3 | 4.7 +/- 1.2 | 3.8 +/- 0.6 | 0.28 +/- 0.05 |
| M4 | 1.4 +/- 1.1 | 2.8 +/- 0.5 | 0.28 +/- 0.05 |
| M5 | 3.2 +/- 1.1 | 3.0 +/- 0.5 | 0.28 +/- 0.05 |

*Table 1: Relative sea level trends (mm/yr) recorded by the M3, M4 and M5 tide gauges (estimated with and without the seasonal cycles) as well as the GNSS-based vertical land motion (mm/yr) at the Senetosa site.*

The M4 time series displays several gaps over the study period. In addition, the record (seasonal cycle not removed, Fig. 12a) shows a large positive anomaly in 2015, not seen by M3 neither M5. M3 has also a large gap in 2009/2010, as well as other gaps 2012 and at the end of the record. A suspect drop is also visible in 2005 on Fig. 12b (seasonal cycle removed). Thus the M5 record seems the most reliable, even if the trends from M3 and M4 are close to M5 (see Table 1). The computed (relative) sea level trend (uncorrected for the VLM) is on the order of 2.8-3.8 mm/yr over the study period (seasonal cycle removed). If the GNSS VLM trend is accounted for, this range becomes 3.1-4.1 mm/yr. This value is significantly less than the altimetry-based sea level trends reported here in the last 4-5 km to the coast. On the other hand, the tide gauge trend agree well with the altimetry-based trends reported at distances greater than > 4 km from coast. While the reported altimetry-based sea level trend increase may disqualify our retracked sea level data in the vicinity of the coast, in


the next section we discuss the possibility that some coastal processes affect sea level in a
band of a few km from the coast while being attenuated very close to the shore where the tide
gauges (in particular M5) are located. .

## 6. Small scale coastal processes

Compared to deep-ocean sea level, sea level close to the coast can be impacted by various
small-scales processes resulting from the morphology of the coastline, the depth of the
continental shelf, the presence of a river estuary, etc. (Woodworth et al., 2019). Thus coastal
sea level may significantly differ from open ocean sea level over a large range of temporal
scales. In terms of trends, the open ocean sea level essentially results from processes affecting
the global mean sea level (mean ocean thermal expansion, land ice melt and land water
storage changes) (e.g., WCRP, 2018) and the superimposed regional variability (regional
changes in ocean thermal expansion, atmospheric loading and fingerprints due to the solid
Earth response to changing ice mass loads; Stammer et al., 2013). At the coast, in addition of
these two contributions, local variations in other processes may cause additional small-scale
sea level changes at interannual to decadal time scales, such as trapped Kelvin waves,
upwelling/downwelling effects, eddies, wind-generated waves and swells, shelf currents, water
density changes related with river runoff in estuaries (see Woodworth et al., 2019 for a detailed
discussion on forcing factors affecting sea level changes at the coast). Note that we do not
discuss vertical land motion here since our objective is to understand the observed change
in 'geocentric' sea level as measured by satellite altimetry.
In the case of Senetosa, river runoff and trapped Kelvin waves are not supposed to affect
coastal sea level. Could other processes like trends in wind-generated waves and coastal
currents explain the slow increase in sea level trend towards the coast? These are discussed
below.

### 6.1 Effect of waves on SLA and SSB

We first discuss the effect of waves. The contribution of wind-generated waves to coastal sea
level changes has been investigated in a number of recent studies (e.g., Melet et al., 2018,
Dodet et al., 2019). As thoroughly discussed in Dodet et al. (2019), wind-generated waves
have the capability to significantly change sea level variations at the coast, even at the time
scales of interest here. The shoaling and breaking of waves in the shelf shallow waters raises
the mean water level in the so-called near-shore and surf zones (last ~1 km to coast), a

process called wave set-up. Wave set-up is proportional to offshore significant wave height,
and if the latter displays a temporal trend due to a trend in wind forcing, it may cause a sea
level trend in the coastal zone.
The relationship between offshore wave height and wave set-up is known empirically only
(Dodet et al., 2019). To first order, wave set-up is related to offshore SWH, wave period and
beach slope. The bathymetric profile along the Jason track 85 (from 45 km offshore to coast)
is shown in Fig. 13. We note an abrupt increase of more than 500 m in the last 5 km to coast,
corresponding to a slope of 0.1.

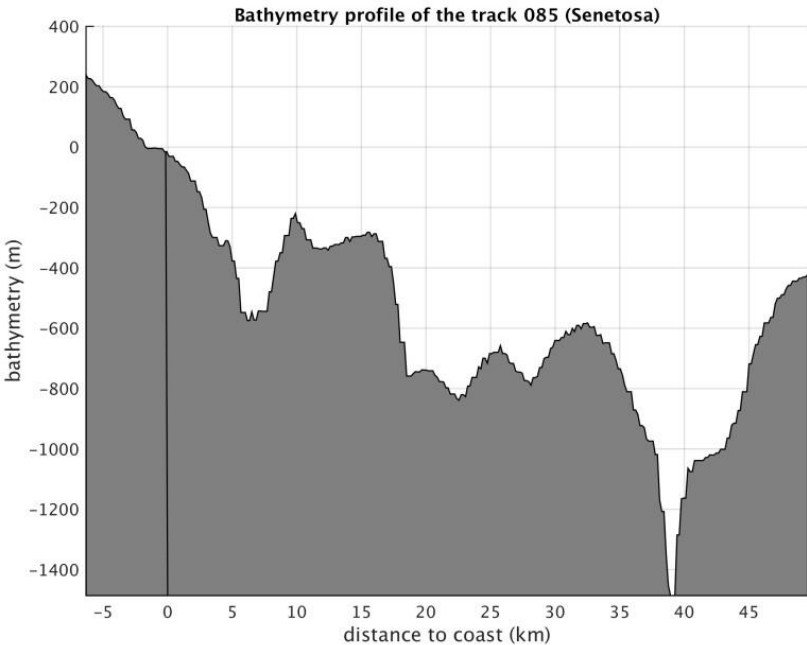

*Fig. 13: Bathymetric profile (meters) along Jason track 85 from 45 km offshore to coast*

If the bathymetric slope near Senetosa is known, it is not the case for other parameters
involved in the relationship between SWH and wave set-up. This is the case in particular for
beach soil  characteristics, sediment size, etc. A  large variety  of formulations have been
proposed for this relationship, based on in situ observations collected at different coastal sites
(e.g., Dodet et al., 2019). However, these are not necessarily applicable to our study case as
some local beach parameters are not known. But it is generally assumed that wave set up does
not exceed 20% of SWH. Thus, as a preliminary approach, we analyzed offshore SWH data
only, in order to highlight their temporal variability over our study time span.
For  that  purpose  we  considered  wave  field  data  from  the  ERA5  reanalysis
(https://www.ecmwf.int/en/forecasts/datasets/reanalysis-datasets/era5). The ERA5 reanalysis

provides gridded SWH time series at monthly interval, from 1979-present, thus covering our
study period. The grid size resolution is 0.5 degree. Using this data set, we computed 2-D
SWH trends over 2002-2016, shown in Fig. 14. We note high positive wave height trends
west of Corsica and Sardinia over this period. Along the Jason track 85, in the vicinity of
Senetosa, the trend is on the order of 5 mm/yr. Note that we also computed the wind trend using
the same ERA5 reanalysis gridded data over the same period (2002-2016). The map (not shown)
displays positive trends in wind south of Corsica, although with smaller amplitude than along
the western coast of Sardinia, like the wave height map shown in Fig.14.
From the above discussion, we deduce that wave set up would not contribute by more that 1
mm/yr to the coastal sea level trend. Noting in addition that wave set up would affect sea level
in the close vicinity of the coast only (i.e., not over 4-5 km distance, X. Bertin, and J. Wolf,
personal communications), it is very unlikely that wave set up can explain the reported coastal
sea level trend.

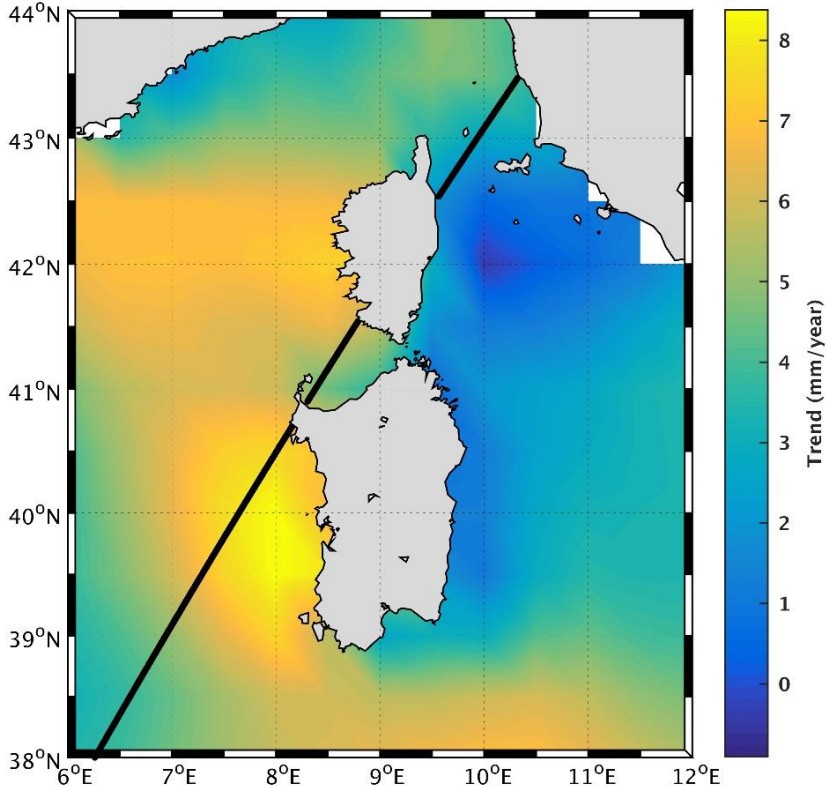


*Fig. 14: Wave height trends (in mm/yr) over 2002-2016 in the western Mediterranean Sea*
*(data from ERA5 reanalysis)*

We further investigated the effect of waves on the ssb correction, hence on SLAs. For that
purpose, we computed the correlation between wave height time series and difference in sea
level between each 20 Hz altimetry point and a reference altimetry point located in the
open ocean (chosen here at 15 km from the coast). We consider differences in sea level
anomalies in order to remove the common o c e a n signal affecting sea level close to the
coast and offshore, e.g., the global mean sea level rise and its superimposed regional
variability. By computing the sea level differences between 15 km offshore and
coast, the latter large-scale sea level components are removed, leaving only
small-scale signals occurring very close to the coast. Data from the ERA5 grid
closest to Senetosa were used (the center of the considered grid point is located at 24 km from
the first valid point on the Jason track and 25 km from Senetosa). The correlation values are
shown in Fig. 15 against distance to the coast. From a distance of ~3 km from the coast towards
the deep sea, the correlation between wave height and sea level difference is insignificant while
it clearly increases from ~3 km to the coast. This suggests that there is a link between the
variations in waves and SLA variations in the 0-3 km domain close to land.
We performed the same analysis but now using the M5 tide gauge record as reference (the M3
tide gauge record having too many data gaps). This is also shown in Fig. 15. Surprisingly, we
find exactly the same behavior of the correlation coefficient, i.e., no correlation offshore
(points located at distance > 3 km from coast) and an increase in correlation in the last 3 km
to the coast. This suggests that waves may affect SLA only in the domain 0-3 km from coast
but that at the tide gauge site, waves have no influence. Obviously, this could be via the ssb
correction applied to SLA data.

**Correlation between SWH and Sea Level differences**



*Fig. 15: Correlation between the wave height (SWH) time series (from ERA5 grid mesh close to Senetosa) and altimetry-based sea level difference time series between every 20 Hz point and a reference point. (a) The reference time series corresponds to a point located at 15 km from the coast. (b) The reference time series is the M5 tide gauge record.*

It has been demonstrated that applying the ssb correction to altimetry data, in particular to high-frequency data as in this study, reduces the correlation between SWH and range (and, consequently, SLA) (Passaro et al., 2018). The ssb correction is mainly a function of SWH: it removes from the range estimation an effect that is directly proportional to the wave height. This means that if this ssb correction is not applied, it has to be expected that the SLA record will be correlated with the SWH record. To illustrate this somewhat differently, Fig. 16a shows wave height time series superimposed to altimetry-based difference in SLA time series (reference point at 15 km, as in Fig. 15) for a few points located in the 0-3 km domain close to the coast and an additional point located farther from the coast. Here again, data from the ERA5 grid closest to Senetosa have been considered for the calculation. The correlation between SWH and difference SLA time series is indicated on each plot. We clearly see that it is significant only for points close to the coast. Distant offshore points do not show such a correlation. Although the correlation is dominated by the seasonal signal, Fig. 16a shows the two time series are also correlated at interannual time scales.

We argue that when the range close to the coast is not being properly corrected for the ssb, this results in a correlation between ssb and SWH. To verify this, we repeated this correlation analysis but now using the ssb correction (from both the ALES and MLE4 retrackings) instead of the SLA differences. The corresponding figure is shown in Fig. 16b.

(a)

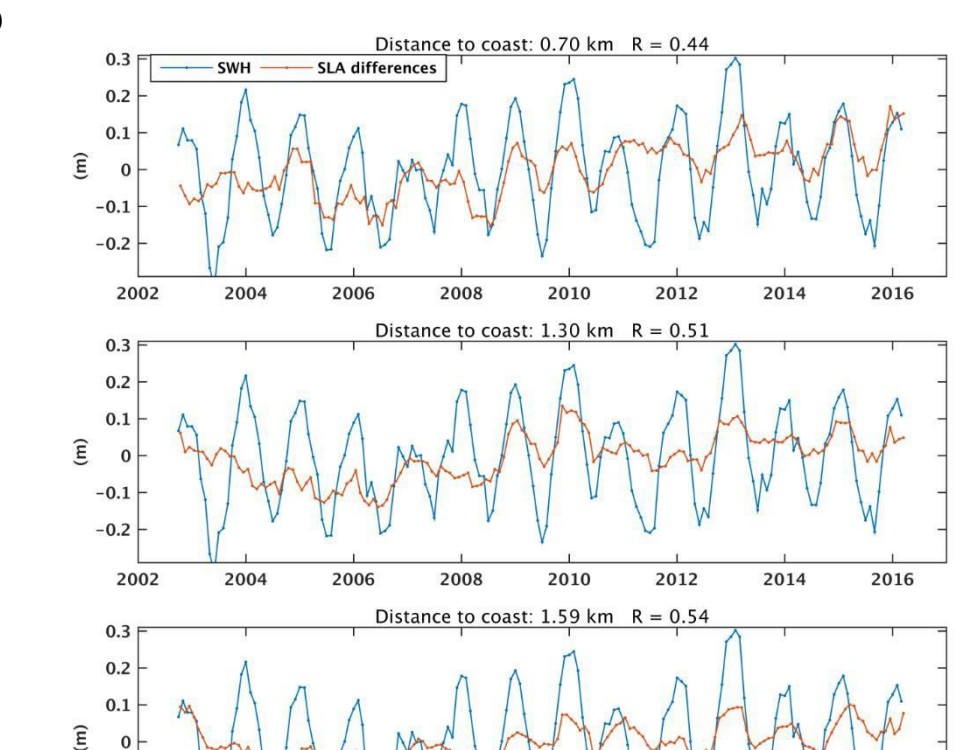

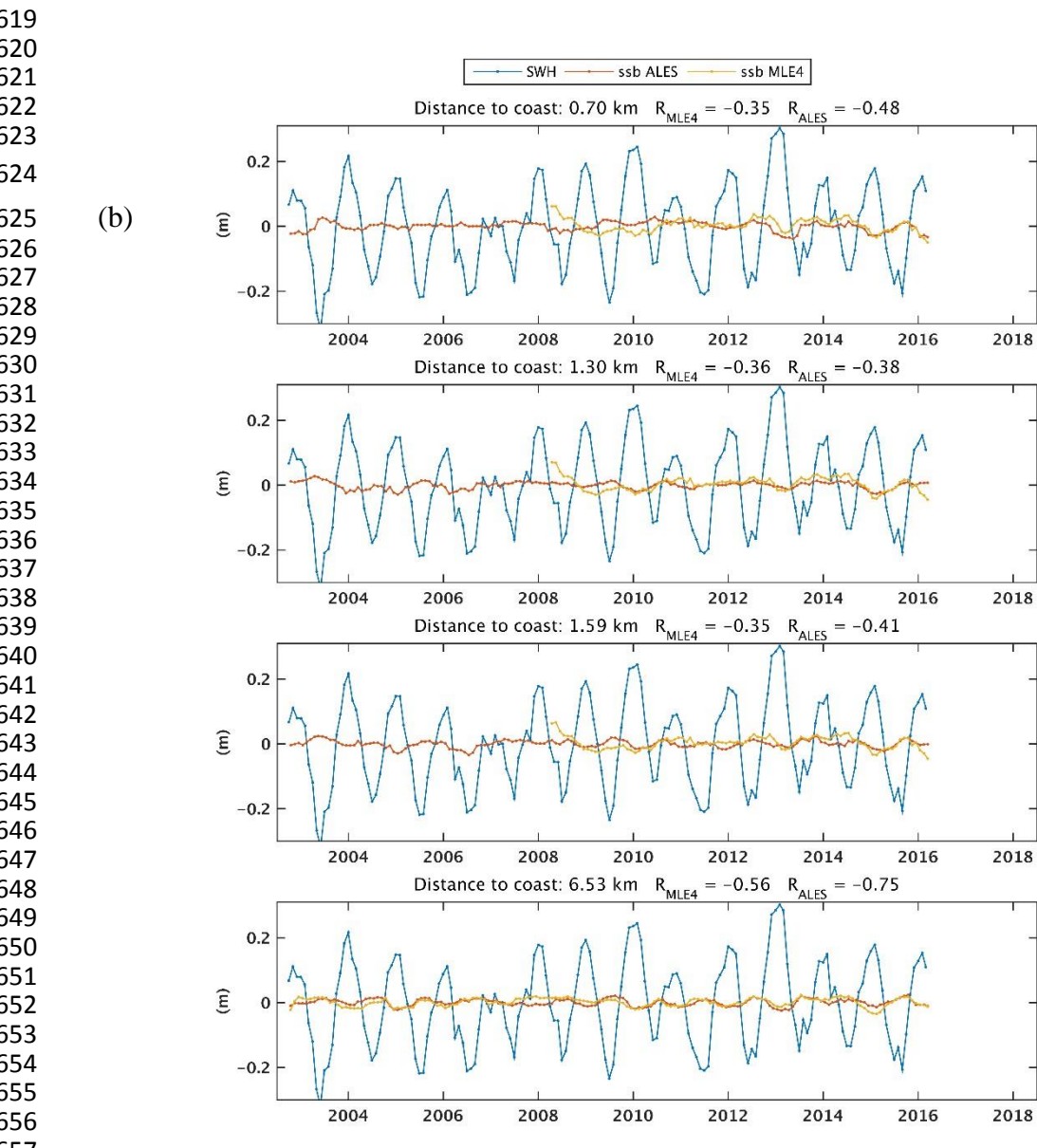

*Fig. 16: (a) Time series of ERA5-based wave height time series (blue curve) and of altimetry-based SLA differences (orange curve) between 20 Hz points at different distances from coast (indicated on each plot) and a reference point (located at 15 km). (b) same as (a) but using ALES ssb instead of SLA differences. On Fig. 16b, MLE4 ssb are also shown for the Jason-2 time span (yellow curve). R is the correlation coefficient.*

As expected, ssb is correlated with SWH away from the coast, but the correlation decreases in the last few km to the coast, suggesting that the relationship used to express the link between ssb and SWH is less adapted in the coastal domain than in the open sea, either because of change of wave properties (which makes the ssb model invalid) or because of a wrong estimation of SWH very close to the coast. This is also illustrated in Fig. 17 that shows the correlation between ssb and SWH against to distance to the coast (for both ALES ssb and MLE4 ssb). Between 1 km

and 4 km, the correlation between SWH and ssb decreases. It is worth noting however that the correlation remains higher for ALES ssb than for MLE4 ssb.

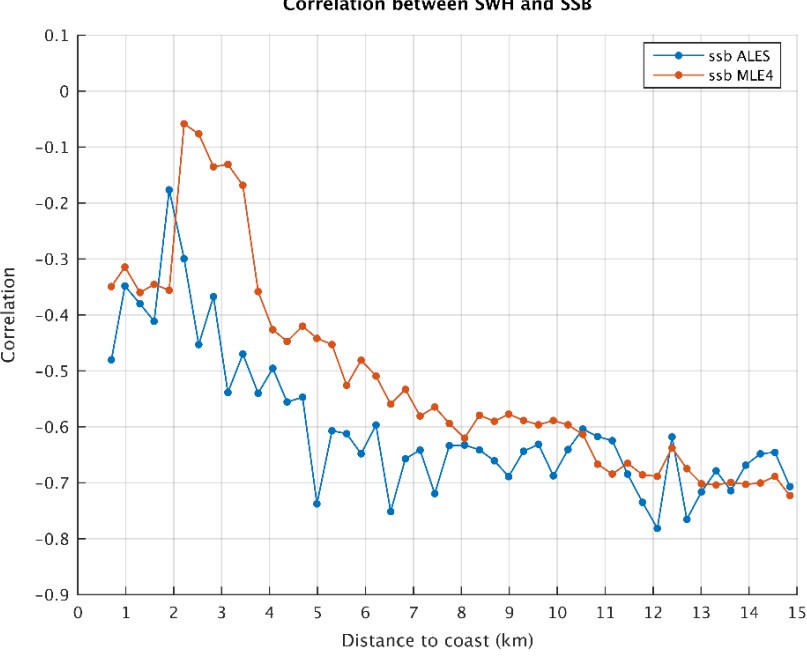

Fig. 17: *Correlation between significant wave height (SWH) time series and ssb time series between every 20 Hz point and a reference point.*

We conclude from these tests, that the correlation between SLA and wave height at 20 Hz points close to the coast is very likely due to imperfect ssb correction. Thus we can now exclude any direct effect of waves (e.g., trend in wave set-up) as a candidate to explain the SLA trend increase close to the coast. Are the reported SLA trends in the last few km to the coast due to inadequate formulation of the relationship between SWH and ssb as the satellite approaches the coast remains so far an open question. While we cannot exclude that the ssb correction is imperfect close to the coast, it seems unlikely that it would produce such large trends as those observed in the SLAs.

## 6.2. Effect of coastal currents and comparison with an ocean model

In this section we briefly address the effect of coastal currents on the SLAs. There are only few published studies on the circulation in the Senetosa region (e.g., Bruschi et al., 1981, Manzena et al., 1985, Cucco et al., 2012, Gerigny et al., 2015, Sciascia et al., 2019). These indicate that the dominant characteristics of the circulation in the Corsica channel (Bonifacio Straits) is a flow predominantly directed northward from the Tyrrhenian Sea to the Ligurian


Sea and that the water motion is mainly wind-driven. The study by Gerigny et al. (2015)
based on in situ measurements collected during a cruise in 2012 and use of a high-resolution
regional hydrodynamic model (MARS3D) shows that the circulation is mostly wind-driven,
forced by westerly winds half of the year and strong easterly winds in winter, generating strong
local currents and mesoscale structures in the western part of the channel. We have
downloaded the currents data generated by the MARS3D model, a coastal hydrodynamical
model developped by IFREMER (Institut Français de Recherche pour l'Exploitation de la
Mer; Lazure and Dumas 2008). There is a high-resolution (400 m) version available for the
Corsica region, for the years 2014 to present
(http://www.ifremer.fr/docmars/html/doc.basic.intro.html). The model does not assimilate
altimetry data nor any other type of data. Because this dataset has only 2.5 years of overlap with
our study period, we cannot compute trends. However, to gain some insight on the circulation
configuration, we examined the currents pattern over the year 2014. In agreement with the
literature, we observed a strong zonal current during the winter months close to Senetosa. An
example of the zonal component of the barotropic current south of Corsica is shown in Fig.18
for January 2014. We note a clear westward current along the Senetosa coast starting at ~4 km
from the coast. It is also worth noting that it does not extend to the shoreline, thus may not
influence tide gauge measurements.

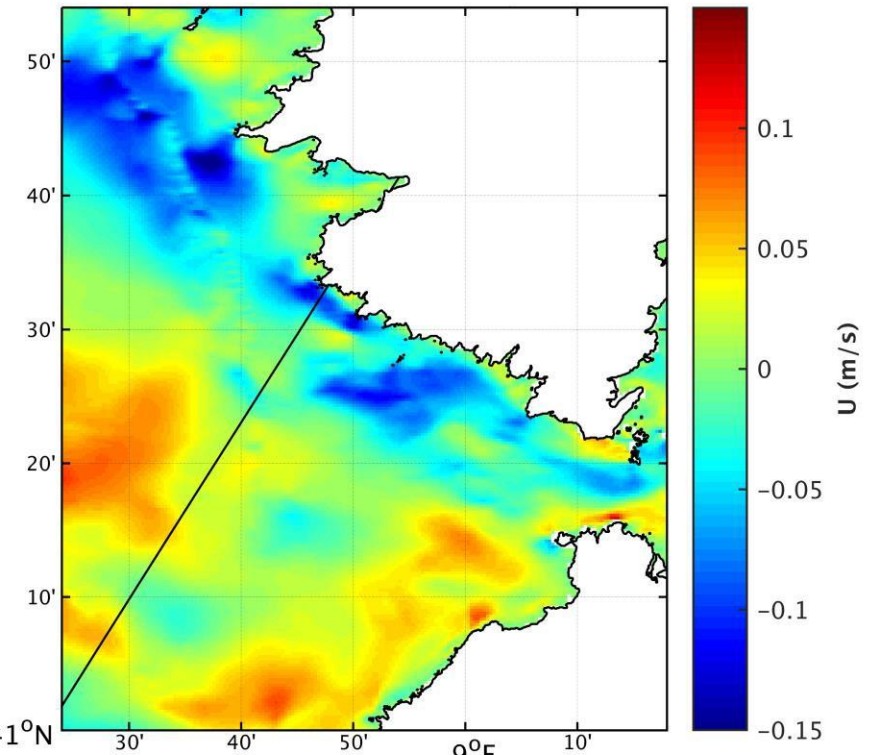


*Fig.18: Barotropic current (zonal component) for January 2014 based on the MARS3D hydrographic model. Blue color means westward current. The Jason track (black line) crosses this current at 4 km from the coast.*

We interpolated these current data (for January 2014) along the Jason track. This is shown in Fig.19 against distance to the coast. The current intensity is close to zero at distances >5km from the coast. In the last 5 km to the coast, there is a steep intensity increase, exactly over the same distance range as the SLA trend increase. Since the model resolution is ~400m, i.e., about the same resolution as the 20 Hz along-track SLAs, we find this result highly promising.

**Zonal current velocity along track**

*Fig.19: Barotropic current (zonal component) for January 2014 based on the MARS3D hydrographic model interpolated along the Jason track, as a function of distance to the coast. Negative values mean westward current.*

Of course, we cannot extrapolate backward in time nor offer any solid conclusion so far. But we cannot exclude that the observed sea level trend increase is linked to an increase in intensity of this winter current during our study period. This obviously will need much deeper investigation, at least over the time span of availability of the model data.

## 7. Conclusion

In this study, we have investigated the differences between coastal and deep ocean sea level changes at the Senetosa site, using new ALES-based retracked sea level data from the Jason-1 and Jason-2 missions. We indeed observe a slow increase in sea level trend at short (< ~4-5 km) distance from the coast compared to offshore. A series of test shows that this behavior does not result from artifacts due to spurious trends in the geophysical corrections applied to the altimetry data, decreasing percentage of valid data, or errors in the intermission bias nor errors in range estimates due to distorted radar waveforms.

While the paper was in review, an update of the results presented above has been recently performed extending the SLA time series with Jason-3 data up to June 2018 (coastal trends based on Jason-1, 2 and 3 over 2002-2018 at several hundreds of coastal sites located in six different regions worldwide are presented elsewhere; The Climate change Initiative Coastal Sea Level Team, 2020). Although the coastal trends within the 2-3 km to the coast are slightly lower than those reported above, exactly the same behavior is found, as shown in Fig.20 that compares coastal trends over 2002-2016 and 2002-2018. Thus, the trend increase close to the coast observed at Senetosa is not due to the limited length of the time series, although its amplitude decreases as the record length increases. Similarly, the geophysical correction trends present the same behavior over both time spans. It is worth mentioning that in the extended study (2002-2018), among the 429 studied coastal sites, coastal trends do not in general differ from open ocean trends (within +/- 1 mm/yr), except at a few sites (The Climate Change Initiative Coastal Sea Level Team, 2020), Senetosa is one of them. This is why we made a focus on that particular site.



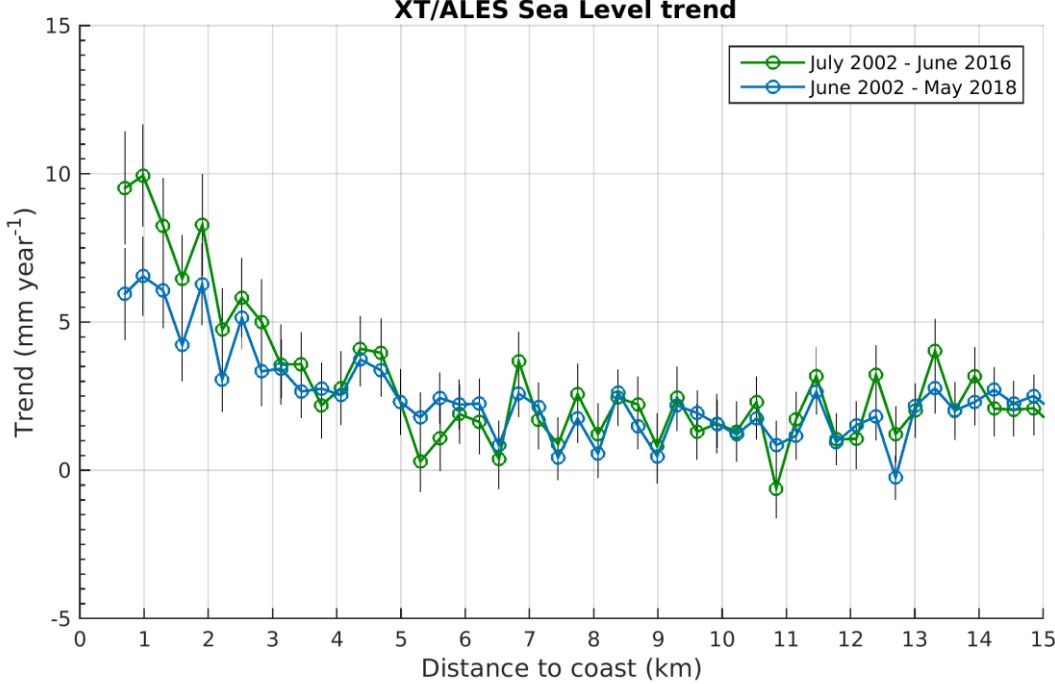

*Fig.20. Altimetry-based sea level trends at Senetosa, over two periods: (1) July 2002-June*
*2016, green curve and (2) June 2002-May 2018, blue curve. Black vertical bars*
*correspond to trend uncertainties.*

Among the physical mechanisms able to explain the coastal trend increase in the study region,
we have first explored waves, then currents. We investigated the wave effect on sea level
along the Jason track and found that wave set up has a too small magnitude and is localized
too close to the shore to explain the observed continuous SLA trend increase in the last 4-5
km to the coast. On the other hand, the correlation reported between altimetry-based SLAs
and SWH very likely results from the imperfect ssb correction applied to the data.
Nevertheless, if less accurate in the coast vicinity, the ssb trend seems unable to explain the
reported SLA trend increase. We next investigated the effect of coastal currents. Using the
MARS3D high resolution model developed by IFREMER for coastal studies, we noted the
presence of a winter current along the Senetosa coastline. Projection of this current along the
Jason track (for January 2014) shows a steep increase in intensity over exactly the  same
distance to the coast as the SLA trend increase. This may be an indication of a current-related
origin. More studies are definitely needed to confirm  the results presented here.  However, if
further investigations confirm the effect of currents, it will be a demonstration  that small-
scale processes acting in the vicinity of the coast may have the capability to make  coastal sea
level changes drastically different from what we measure offshore with classical  altimetry.

**Acknowledgements**

This study is a contribution to the ESA Climate Change Initiative (CCI+) Sea Level project. Yvan Gouzenes is supported by an engineer grant in the context of this project (ESA SL_cci+ contract number 4000126561/19/I-NB). We thank a number of colleagues for very fruitful discussions on the effect of waves on tide gauges and coastal sea level, in particular (by alphabetic order) Angel Amores, Xavier Bertin, Svetlana Jevrejeva, Goneri Le Cozannet, Marta Marcos, Judy Wolf and Phil Woodworth. We also thank two anonymous reviewers and the Editor for their comments and suggestions to improve the manuscript.

**Data availability**

The coastal sea level data analyzed in this study are available from the Nature Scientific Data article (The Climate change Initiative Coastal Sea Level Team, 2020). The ERA wave field data from the ERA5 reanalysis are available from the following web site: https://www.ecmwf.int/en/forecasts/datasets/reanalysis-datasets/era5.
The MARS3D model can be downloaded from the web site: http://www.ifremer.fr/docmars/html/doc.basic.intro.html)

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

**RESPONSES TO THE EDITOR'S COMMENTS (*in italics*)**

**Topic Editor Decision: Publish subject to minor revisions (review by editor)** (31 Jul 2020) by Joanne
Williams
Comments to the Author:
Dear authors,
Thank-you for your revised manuscript, and response to the reviews. There are a few minor issues
remaining, as follows.
Table 1: please be consistent in usage of Ku-band (ideally $K_u$)
*Corrected*
Table 1: sea-state bias (spelling)
*Corrected*
line 159: western Mediterranean Sea
*Corrected*
line 167: "Senetosa is operating"? The place can't operate it. Perhaps "Since 1998 a calibration site has
operated near the Senetosa lighthouse, with support from..."?
*Modified as suggested*
line 176: You could tidy this a little, eg "M4/M5 as a few cm apart, sheltered from the northwest-ward
wind, whilst M3 is a 1.7km eastward, more exposed to open-sea conditions. "
*Modified as suggested*
Fig 1: Ensure the copyright is correctly handled for the Google earth image - See https://www.ocean-
science.net/for_authors/manuscript_preparation.html
*OK*
line 210: Outliers are omitted in computing the regression line.
*Corrected*
line 245, 250 (and check elsewhere): "Figure 4 shows..." or "as shown in Fig. 4"
*Corrected*
line 255: Grammar. I suggest: "In coastal areas, precision of sea surface height from altimetry is limited
by inaccuracies in some of the applied geophysical corrections (including sea state bias, wet
tropospheric correction, dynamical atmospheric correction and ocean tides) and from the distorted
shape of the radar waveforms as the satellite approaches land (Vignudelli et al., 2011 and Cipollini et al.,
1051   2018).
*Modified as suggested*
line 260: Bit vague. I think you mean that data that is not flagged can still have errors due to coastal
proximity?
*Modified*

line 265/269 : Try not to switch tense. examined?

*Modified*

line 269: reads like d(ssb)/d(distance to coast). I suggest "We plotted trends in geophysical correction
against distance to the coast, for sea-state bias..."

*Modified as suggested*


FIg 6b is unnecessary. Suggest line 329 becomes "We resampled the along-track sea-level records
keeping only the 80% of data common to all along track positions at a given time."

*Fig.6b has been deleted*
*Line 329 modified as suggested*

line 333 cut "then"

*Done*

Fig 7: as a general rule for color-vision accessibility, avoid red vs green lines. Replot if possible, however
it does not significantly affect the message of this figure so I don't insist on it. However in Fig 20 it does
matter, please replot.

*We have not modified the figures except Fig.20 which has been redrawn (blue instead of red)*

line 383: full stop.

*Done*

line 515: cut "well"

*Done*

Fig 10: would be more intuitive rotated to align with the map. However it is not essential.

*In effect, this is not essential*

Fig 12: It would have been good to see the altimetry results alongside here.

*It will be the purpose of another paper in preparation where we compare tide gauge records with*
*coastal SLA time series at Senetosa and several other sites*

line 595: "it is very unlikely that wave set up explains the reported coastal sea level trend."

*Modified as suggested*

lines 604-715: The argument in this section is rambling and hard to follow. It needs tidying up and
probably condensing. You can simplify a lot of sentences.

As I understand your argument ...
Could waves explain the SLA trends approaching the coast? SWH at a nearby ERA grid point has little
correlation with SLA at 15km, but has some correlation with SLA at the coast, Fig15 & Fig 16a. One
mechanism for waves to affect the altimetry corrections is via ssb. SWH at the ERA grid point does not
correlate well with ssb near the coast (Fig 16b & Fig 17). So it is possible the wrong ssb is used as an
altimetry correction near the coast (slightly better in ALES). Therefore though the waves do correlate
with SLA, it's not via ssb, so it's not causal. So despite fig 16a, the argument in the first part of 6.1 still
holds? You say you have eliminated waves as the explanation.
However in section 6.2 we learn that the winds are highly seasonal and affect the local currents.
Seasonal changes in wind direction would directly affect the local SWH, since sometimes the coast will
be sheltered. More localised wind and wave information would be very helpful here. Otherwise I think
you can only say that there is no strong evidence for waves and ssb causing the trend in SLA.
If I've got this right it's after a lot of unpicking. Please clarify. If I've got it wrong, then this *really* needs
clarifying.
*It seems that our text was unclear because the above comment is exactly the opposite of what we*
*wanted to say, ie., that the correlation between waves and SLA is do to the wrong SSB.*
*Schematically:*
*1) SLA without SSB correction is always correlated with SWH (the SSB phenomenon is a*
*function that is directly proportional to SWH. It is THERE until we remove it with a correction)*
*2) If we apply the SSB correction correctly, we decrease the correlation between SLA and SWH*
*3) If the SSB correction is wrong, the correlation remains. This is what we see very close to the*
*coast.*
*The confusion comes from the fact that, although we are focusing on the explanation that the ssb*
*correction is not being effective very very close to the coast, we still (during the section) keep*
*open the possible explanation that the correlation between SWH and SLA is physical ("This*
*clearly suggests that computed SLAs are impacted by waves in the last few km to the coast on a*
*broad range of time scales").*
*We have modified the text of the revised version to make the argumentation  clearer. We have also*
*added a short introduction to explain what we want to demonstrate.*
specifics from this section:
line 604: cut "However"
*Done*
line 609: we consider sea-level anomalies...
*Corrected*
line 610: cut "mesh"
*Done*
line 610: Please indicate the position of the ERA5 grid point you use on one of the maps. ERA5 grid
spacing is about 30km so it matters. It is a pity you don't have anything finer resolution here.
*We have added a sentence in the text indicating the position of the closest  grid mesh to Senetosa*
line 638-640: No. Fig 16a doesn't show the interannual correlation, because it is swamped by the
seasonal signal. You'll have to replot this or at least recalculate R after filtering the seasonal out.
*The correlation is mostly based on the seasonal signal but as the amplitude of the latter varies with time*
*in the same way in both time series, this indicates that the correlation also holds at interannual time*
*scale*
line 693: The correlations are really rather weak here. Suggest "As expected, ssb is correlated with wave
height away from the coast, but..." It is difficult to see any relationship from the figure. Replot with the
orange ssb curves on a different scale (eg use a right-hand axes).
*Modified as suggested*
line 699: You haven't calculated significance. Rephrase.
*Modified as suggested*
lines 729-733: could condense this.
*Done*
Fig 18 & 19: could be combined. And why zonal only? You need the meridional component too to give
the along-shore current.
*Only the zonal components shows some trend. More investigation is on going using the MARS3D model*
*around Senetosa and this will be the subject of another paper in preparation.*
Please indicate 15 km along-track distance on at least one map.
*Done in the text*
Although you can't derive a trend from the MARS3d model, and it's fair enough to pass on the coastal
oceanography to another study, it would be very helpful to do a seasonal comparison, perhaps plot a
different season in Fig 18.
*See above comment*
line 832: cut "elongated"
*Done*
line 834: an indication
*Corrected*
You have not met all requirements in
https://www.ocean-science.net/for_authors/manuscript_preparation.html
Please attend particularly to data referencing, but check through other requirements as well. Don't rely
on the typesetter to find them!
The MARS3d model, ERA5 data (they have a standard format statement required), altimetry data all
need correct acknowledgement and statement of where to access the data. This should be in a Data
Availability section.
*A Data availability section added*