# Peer review of "Coastal Sea Level rise at Senetosa (Corsica) during the Jason altimetry missions"

_Ocean Science, 2020_

## Referee Comment (RC1) · Anonymous Referee #1 · 27 Mar 2020

Review of the paper: os-2020-3 Title: Coastal Sea Level rise at Senetosa (Corsica) during the Jason altimetry missions

General assessment This paper addresses a relevant topic of research, the determination of local coastal sea level trends from satellite altimetry. The study focuses on a Jason track crossing Corsica Island at the Senetosa site. The analysed period spans 14-years (from July 2002 to June 2016). Altimeter data used include Jason-1 and Jason-2 20Hz measurements, ranges from the ALES retracker and corrections from the X-TRACK system. The main conclusion of the paper is that, provided altimeter-derived coastal sea level trends are reliable, these trends can be significantly different from the corresponding open ocean trends. Most effort is put in justifying that the results are not due to e.g., spurious trends in the geophysical corrections, imperfect intermission bias

estimate, decrease of valid data close to the coast and errors in waveform retracking. The paper is scientifically sound, generally well written and structured. The paper can be accepted subject to minor revision. A few suggestions are given to improve paper clarity.

Detailed comments: 1) MY major suggestion to authors is to include a discussion about the connection (or not) between the results in this particular site with the global results of the CCI project. Did they find many sites where coastal sea level is significantly different from that offshore? Is this a representative site or an exceptional result?

2) Since the main focus of the paper is trying to discard causes that might explain the observed trends, it is important to give enough detail on the altimeter data used and adopted processing, so that the reader can follow the discussion with enough information. For example, saying that that corrections are those adopted in the X-TRACK system is not enough. At least the corrections that most affect coastal sea level, in addition to the SSB, the wet tropospheric correction and ocean tides should be discussed in more detail. Information should be given, with appropriate references, on: i) models used (e.g., original wet tropospheric correction from the Jason GDRs (MPA algorithm from Brown ,TGARS 2010) or from GPD (Fernandes, RSE 2015)?; ii) tide model from FES2014 or any other model? How big are tides in this site?; iii) rate at which each of these corrections is provided (1Hz or 20Hz)? In case of 1Hz corrections interpolated to 20Hz, they don't have enough detail to cause differences in trends at scales of few km, discussed in this paper.

Fig. 2: The grey square is hardly visible. Please improve.

Section 4.1: please explain how the standard deviation of trends is computed.

Section 4.2. A more recent reference on coastal altimetry than that by Vignudelli et al., 2011 is the book chapter "Satellite altimetry in coastal regions" by Cipollini et al., 2017 in the CRC Press book. Please include.

Section 4.2.1 – "waves could has a" replace by "waves could have a"

---

## Author Comment (AC1) · 12 May 2020

Review of the paper: os-2020-3 Title: Coastal Sea Level rise at Senetosa (Corsica) during the Jason altimetry missions

General assessment This paper addresses a relevant topic of research, the determination of local coastal sea level trends from satellite altimetry. The study focuses on a Jason track crossing Corsica Island at the Senetosa site. The analysed period spans 14-years (from July 2002 to June 2016). Altimeter data used include Jason-1 and Jason-2 20Hz measurements, ranges from the ALES retracker and corrections from the X-TRACK system. The main conclusion of the paper is that, provided altimeter-derived coastal sea level trends are reliable, these trends can be significantly different from the

corresponding open ocean trends. Most effort is put in justifying that the results are not due to e.g., spurious trends in the geophysical corrections, imperfect intermission bias estimate, decrease of valid data close to the coast and errors in waveform retracking. The paper is scientifically sound, generally well written and structured. The paper can be accepted subject to minor revision. A few suggestions are given to improve paper clarity.

Detailed comments: 1) My major suggestion to authors is to include a discussion about the connection (or not) between the results in this particular site with the global results of the CCI project. Did they find many sites where coastal sea level is significantly different from that offshore? Is this a representative site or an exceptional result? Reponse: This is the objective of another article to be submitted in 2 or 3 weeks to Nature Scientific Data where we present coastal sea level anomalies and coastal sea level trends from the Jason-1, 2 and 3 missions at 429 selected sites (among several thousand studied sites) located in 6 different regions (Northeast Atlantic, Mediterranean Sea, Western Africa, North Indian Ocean, Southeast Asia and Australia) that gives robust coastal trends. It is found that in general, coastal trends do not differ from open ocean trends except at a few sites. Senetosa in the Mediterranean Sea is one of them. We decided to write a separate article on the Senetosa results in which we examine in many details potential errors in the data processing (including spurious geophysical correction) to assess the validity of the observed coastal trend. This kind of detailed analysis is not presented in the Nature Scientific Data paper (impossible to do this at 429 sites in a single article!). In the present revised version, we have added a paragraph in the conclusion section to explain our strategy, mentioning that Senetosa counts among the very few coastal sites where coastal sea level trends differ from open ocean trends.

Added text in the revised manuscript: "An update of the results presented in this paper has been recently performed extending the SLA time series with Jason-3 data up to June 2018 (coastal trends based on Jason-1, 2 and 3 over 2002-2018 at several

hundreds of coastal sites located in six different regions worldwide will be presented elsewhere; The Coastal Sea Level Team, 2020). Although the coastal trends within the 2-3 km to the coast are slightly lower than those reported above, exactly the same behavior is found, as shown in Fig.20 that compares coastal trends over 2002-2016 and 2002-2018. Thus the trend increase close to the coast observed at Senetosa is not due to the limited length of the time series, although its amplitude decreases as the record length increases. Similarly the geophysical correction trends present the same behavior on both time spans. It is worth to mention that in the extended study (2002-2018), among the 400+ studied coastal sites, computed coastal trends do not in general differ from open ocean trends (within 1 mm/yr), except at a few sites. Senetosa is one of them. This is why we made a focus on that particular site." Added figure: "Fig.20. Altimetry-based sea level trends at Senetosa, over two periods: (1) July 2002-June 2016, green curve and (2) June 2002-May 2018, red curve. Black vertical bars correspond to trend uncertainties."

2) Since the main focus of the paper is trying to discard causes that might explain the observed trends, it is important to give enough detail on the altimeter data used and adopted processing, so that the reader can follow the discussion with enough information. For example, saying that that corrections are those adopted in the XTRACK system is not enough. At least the corrections that most affect coastal sea level, in addition to the SSB, the wet tropospheric correction and ocean tides should be discussed in more detail. Information should be given, with appropriate references, on: i) models used (e.g., original wet tropospheric correction from the Jason GDRs (MPA algorithm from Brown ,TGARS 2010) or from GPD (Fernandes, RSE 2015)?; ii) tide model from FES2014 or any other model? How big are tides in this site?; iii) rate at which each of these corrections is provided (1Hz or 20Hz)? In case of 1Hz corrections interpolated to 20Hz, they don't have enough detail to cause differences in trends at scales of few km, discussed in this paper.

Response: In the revised version, we have added a long paragraph discussing the

processing approach and the source of chosen geophysical corrections, explaining why these have been selected for this study. A table dedicated to the geophysical corrections and associated references has been added. Added text in the revised manuscript: "The new X-TRACK/ALES processing system first downloads from the altimetry database hosted by the French National Observations Service for altimetry called CTOH (http://ctoh.legos.obs-mip.fr/), all parameters needed to compute the sea level anomaly (orbit solution, altimeter ranges, instrumental, environmental and geophysical corrections). These parameters come from the Geophysical Data Records (GDRs) data sets distributed by the space agencies for the different altimetry missions. ALES range and SSB products come from TUM. Additional geophysical corrections are provided by the RADS altimeter database (http://rads.tudelft.nl/rads/rads.shtml) and the University of Porto (for the GPD+ wet tropospheric correction, Fernandes et al., 2015). Concerning the geophysical corrections, we used the standards defined in the ESA CCI sea level project (http://www.esa-sealevel-cci.org/). These are summarized in Table 1." Added table : "Table 1" Added text (continued): "A dedicated editing strategy was further applied to eliminate noisy data. For each orbit cycle, the temporal behavior of each geophysical correction was analyzed along the satellite track. Abrupt changes were considered as spurious and removed (Birol el al., 2017). This strategy has proved to be very efficient in recovering a significant amount of valid altimeter measurements that were otherwise flagged in the standard GDR products (Jebri et al., 2016). In a second step, all corrections were recomputed at the 20-Hz high-rate using only the valid data, through interpolation/extrapolation method. The sea level data of each cycle were further projected onto fixed points along a nominal ground track and converted into SLAs by subtracting a reference mean sea surface. At this stage of the processing, a regional dataset of SLA time series with a spatio-temporal resolution of 10 days and 20Hz ($\sim$0.3 km) was produced for each Jason mission. To obtain a single multi-mission product, an inter-mission bias was estimated and removed. This was done at regional level by computing the mean sea level differences between the two missions over their overlapping period (calibration phase). The resulting SLAs were further averaged on a

monthly basis at every 20 Hz point and an additional editing was performed to remove outliers (details in Marti et al., 2019). "

Fig. 2: The grey square is hardly visible. Please improve. Response: The figure has been improved as requested

Section 4.1: please explain how the standard deviation of trends is computed. Response: Done

Section 4.2. A more recent reference on coastal altimetry than that by Vignudelli et al., 2011 is the book chapter "Satellite altimetry in coastal regions" by Cipollini et al., 2017 in the CRC Press book. Please include. Response: Done

Section 4.2.1 – "waves could has a" replace by "waves could have a" Response: Corrected

[Figure]

**Fig. 1.** New Figure 20

[Figure]

**Fig. 2.** New Figure 2

---

## Referee Comment (RC2) · Anonymous Referee #2 · 14 Jul 2020

Jason-1&2 satellite altimetry data, reprocessed (with ALES and X-TRACK) to give fine-resolution along-track sea level time series, including closer to the coast and spanning July 2002 to June 2016, are used to compute coastal sea level trends, specifically where the Jason track crosses Senetosa, southern Corsica. The estimated rate of sea level rise increases in the last 4-5 km to the coast, compared with further offshore; it amounts to about 10 cm extra rise near the coast over the 14 years. Potential altimetry errors are considered. I am not expert in these but the manuscript consideration is convincing. An extra 10 cm rise in such a short distance near the coast appears a lot to attribute to error, despite the extra near-coast trend being absent from coastal tide gauges. The possible contribution of wave set up is also convincingly assessed as too small. The remaining identified possibilities are: the sea-state bias correction

degrades somewhat near the coast – certainly SLA difference (near-coast minus 15 km offshore) is correlated with SWH; an effect of coastal currents suggested by distribution of winter currents but no mechanism is proposed and the possibility is left for further investigation.

It is rather disappointing that resolution of the origin of the extra near-coast trend is left to further work. It occurs to me to speculate that the nearshore flow might be affecting the surface waves which could give a systematic effect since it is suggested that both occur more strongly in winter, i.e. they are correlated. Even then, there would need to be a trend in intensity of waves and/or current to produce the differential trend in altimetric signal.

The authors might have considered the momentum equation to look at what might cause a trend in differential sea level. My "back of envelope" calculation for geostrophic balance across a flow of strength 0.2 m/s and width 5 km (rather more than considered in the manuscript) gives a difference of only 1 cm. The very narrow shelf suggests that wind-forced set-up will be very small.

Ultimately, there is a question still unanswered but thereby needing publicity in order to make scientific progress, especially as there might yet be serious implications for near-coast altimetry.

In general the work is well presented and the intended meaning is expressed clearly. On this count, one "technical" matter concerns lines 111-112; Senetosa is "southern Corsica" rather than "south of Corsica" and nearer 9E ? Some explanation of the different colours of the sea in figure 2 might be useful.

---

## Author Comment (AC2) · 23 Jul 2020

Responses to Reviewer 2's Comments

Comments: Jason-1&2 satellite altimetry data, reprocessed (with ALES and X-TRACK) to give fineresolution along-track sea level time series, including closer to the coast and spanning July 2002 to June 2016, are used to compute coastal sea level trends, specifically where the Jason track crosses Senetosa, southern Corsica. The estimated rate of sea level rise increases in the last 4-5 km to the coast, compared with further offshore; it amounts to about 10 cm extra rise near the coast over the 14 years. Potential altimetry errors are considered. I am not expert in these but the manuscript consideration is convincing. An extra 10 cm rise in such a short distance near the coast appears a lot

to attribute to error, despite the extra near-coast trend being absent from coastal tide gauges. The possible contribution of wave set up is also convincingly assessed as too small. The remaining identified possibilities are: the sea-state bias correction degrades somewhat near the coast – certainly SLA difference (near-coast minus 15 km offshore) is correlated with SWH; an effect of coastal currents suggested by distribution of winter currents but no mechanism is proposed and the possibility is left for further investigation. It is rather disappointing that resolution of the origin of the extra near-coast trend is left to further work. It occurs to me to speculate that the nearshore flow might be affecting the surface waves which could give a systematic effect since it is suggested that both occur more strongly in winter, i.e. they are correlated. Even then, there would need to be a trend in intensity of waves and/or current to produce the differential trend in altimetric signal. The authors might have considered the momentum equation to look at what might cause a trend in differential sea level. My "back of envelope" calculation for geostrophic balance across a flow of strength 0.2 m/s and width 5 km (rather more than considered in the manuscript) gives a difference of only 1 cm. The very narrow shelf suggests that wind-forced set-up will be very small. Ultimately, there is a question still unanswered but thereby needing publicity in order to make scientific progress, especially as there might yet be serious implications for near-coast altimetry. In general the work is well presented and the intended meaning is expressed clearly.

Response: We thank reviewer 2 for his/her comments. Indeed the main purpose of this study was to assess all sources of potential errors of the data processing able to explain the sea level trend increase as the distance to the coast decreases. From the numerous tests performed in this study, none is able to explain the reported trend behavior. We further examined the wave set up mechanism but this process occurs too close to the coast to explain our results. We made the hypothesis that trends in coastal currents could be the process we are looking for. Unfortunately, there is a crucial lack of small scale coastal data, especially in this region, to go farther. Thus we decided to leave the process assessment for further studies, either by our team or by other scientists from the coastal oceanography community. High resolution hydrodynamical

models -if available- may provide some answer.

Comment: On this count, one "technical" matter concerns lines 111-112; Senetosa is "southern Corsica" rather than "south of Corsica" and nearer 9E ? Some explanation of the different colours of the sea in figure 2 might be useful.

Response: Corrected

---

## Author Response (AR3)

**Responses to the Topic Editor**

**Topic Editor Decision: Publish subject to technical corrections** (11 Aug 2020) by Joanne Williams
Comments to the Author:
Thank-you for the revised manuscript. I think it is nearly there, but there a still a few remaining corrections that have not been addressed.

Table 1: sea-state bias not biais!

*Corrected*

line 246 and elsewhere "Figure 4 shows". See https://www.ocean-science.net/for_authors/manuscript_preparation.html .

*Not clear what is meant here. Figure captions modified according to authors guidelines*

Thank-you for the improvements to the text in section 6.1 . I now understand your argument about the ssb correction. Figure 16b still requires improvement. The orange/red curves need to be rescaled to a similar scale to SWH, in order that correlation with the SWH is readily visible. Otherwise you might as well omit it and just quote the correlations, as on the scale provided it shows very little.

*We have removed Fig.16b (almost all co authors are on avacation, including the first author Yvan Gouzenes who prepared the figures. Thus no possibility to redraw it within the requested deadline). Correlations have been quoted in the text.*

Line 574 is not at all clear. 24 km in which direction? Why not just provide a lon/lat? Or as previously suggested, indicate on one of the maps (Fig 18 is probably best). Given the variability in Fig 18, this is relevant.

*The coordinates of the grid cell center have been added in the text.*

Please also indicate 15km along-track distance on Fig 18. It would be very helpful for interpreting this figure alongside the along-track plots. The reader can work it out with Google Maps, but shouldn't need to!

*The 15 km along-track distance has been indicated by the red bar across the Jason track.*

Please cite ERA5 reanalysis data as required by ECMWF: see https://confluence.ecmwf.int/display/CKB/ERA5%3A+data+documentation for details.
Please include both the tide gauges and altimetry data in the data availability section. Provide repository links, preferably with DOIs if possible.

*Done*

And there are still missing elements from https://www.ocean-science.net/for_authors/manuscript_preparation.html . For example, I notice there is no contribution of authors section. Please check this list again thoroughly.

*Added*